# PROMPT OPTIMIZATION ACROSS MULTIPLE AGENTS FOR REPRESENTING DIVERSE HUMAN POPULATIONS

## ABSTRACT

The difficulty and expense of obtaining large-scale human responses make Large Language Models (LLMs) an attractive alternative and a promising proxy for human behavior. However, prior work shows that LLMs often produce homogeneous outputs that fail to capture the rich diversity of human perspectives and behaviors. Thus, rather than trying to capture this diversity with a single LLM agent, we propose a novel framework to construct a set of agents that collectively capture the diversity of a given human population. Each agent is an LLM whose behavior is steered by conditioning on a small set of human demonstrations (task–response pairs) through in-context learning. The central challenge is therefore to select a representative set of LLM agents from the exponentially large space of possible agents. We tackle this selection problem from the lens of submodular optimization. In particular, we develop methods that offer different trade-offs regarding time complexity and performance guarantees. Extensive experiments in crowdsourcing and educational domains demonstrate that our approach constructs agents that more effectively represent human populations compared to baselines. Moreover, behavioral analyses on new tasks show that these agents reproduce the behavior patterns and perspectives of the students and annotators they are designed to represent.

## 1 INTRODUCTION

The growing deployment of Large Language Models (LLMs) as human proxies in research and industry has revealed a critical limitation: these models often produce homogeneous outputs that fail to capture the rich diversity of human perspectives and behaviors (Bao et al., 2024; Wenger & Kenett, 2025; Lee et al., 2024; Lahoti et al., 2023). This issue limits their applicability in domains that require this rich diversity, e.g., NLP tasks such as text paraphrasing (Cegin et al., 2023), simulating human behaviors and opinions in surveys (Xie et al., 2024; Santurkar et al., 2023), evaluating conversational recommendation systems (Yoon et al., 2024), replicating human subject studies (Aher et al., 2023), and simulating diverse roles in educational contexts (Nguyen et al., 2024; Markel et al., 2023; Zhang et al., 2025). In particular, LLMs might not be suitable for statistical inference and pose risks of reinforcing and potentially amplifying prevailing community norms in such domains.

To address these limitations, existing works have explored various approaches to align LLMs with humans. A common approach is to fine-tune a single LLM to align with human preferences, e.g., via Reinforcement Learning from Human Feedback (RLHF) (Ouyang et al., 2022). However, RLHF typically relies on aggregated preferences, which can obscure minority viewpoints and may be unrepresentative of broader populations (Casper et al., 2023; Kirk et al., 2023). Indeed, research suggests that it is fundamentally challenging, if not infeasible, to train one model to simultaneously satisfy a multitude of diverse, potentially conflicting, preferences (Ouyang et al., 2022). Another line of work focuses on test-time alignment, which avoids re-training or fine-tuning and leverages in-context learning capabilities of generative models (Brown et al., 2020). In particular, these approaches adapt model behavior dynamically by "conditioning" LLMs on personas or demographic attributes from humans, using techniques such as in-context impersonation (Salewski et al., 2023), culturally specific prompting (Tao et al., 2024), and demographic-aware prompting (Aher et al., 2023).

In this work, we shift the focus from creating a single agent that represents a human population to constructing a set of representative agents. Figure 1 provides an overview of our approach. Our guiding hypothesis is that a carefully curated ensemble of diverse agents can collectively achieve a

Figure 1: Illustrative example of constructing a set of agents $L$ that is representative of a given human population $\mathcal{H}$. In this example, $\mathcal{H}$ is a group of diverse students working on a set of tasks $\mathcal{T}$ and providing answers. The goal is to create a set of agents $L$ that can accurately represent the students. The resulting agents exhibit different levels of understanding across mathematical concepts, with each agent corresponding to a group of students matched by skill level and task performance.

more faithful representation of a given human population. We approach this by leveraging the power of in-context learning, where an agent's behavior is steered by a small set of behavior-representative demonstration examples provided in its prompt. To this end, we formulate the problem of constructing an optimal set of agents as a joint optimization of their prompts. To make this problem tractable, we cast it as a submodular optimization problem and propose several methods for solving it, offering different trade-offs regarding time complexity and performance. Our experimental evaluation shows that these methods can construct agents representative of different groups of people, and that the agents exhibit behaviors matching those groups on new tasks. In summary, our main contributions are:

- We propose a novel formulation that casts the construction of a representative set of LLM agents for a diverse human population as a submodular optimization problem (§3).
- We instantiate this submodular optimization framework and propose methods for selecting sets of representative agents, offering trade-offs between computational tractability and performance (§4).
- We empirically demonstrate the efficiency of our approach in constructing agents that capture the behavior of a given human population in educational and crowdsourcing settings (§5).
- We conduct behavioral analysis on new tasks and show that agents constructed by our methods exhibit behaviors similar to the students and annotators they are intended to represent (§5).

## 2 RELATED WORK

**LLMs for crowdsourcing and simulating human behaviors.** Large language models have emerged as tools for facilitating crowdsourcing tasks and simulating human behaviors. The applications of these models have expanded across multiple domains, enabling research at speed and scale while reducing potential risks to human crowdworkers. Recent studies have demonstrated LLMs' effectiveness as virtual crowd workers for various Natural Language Processing tasks, including data annotation (Moskovskiy et al., 2024), text paraphrasing (Cegin et al., 2023), named entity recognition and relation extraction (Zhang et al., 2023), and text classification (Sun et al., 2023). Another line of work investigated LLMs' capacity for simulating human behaviors and opinions (Xie et al., 2024; Santurkar et al., 2023), evaluating conversational recommendation systems (Yoon et al., 2024), replicating human subject studies (Aher et al., 2023), and simulating diverse roles in educational contexts (Nguyen et al., 2024; Markel et al., 2023; Zhang et al., 2025).

**Diversity and biases and in LLMs outputs**. Current LLMs produce outputs with low diversity and biases, making them unrepresentative of many population groups. Studies have revealed systematic homogeneity in LLM responses (Yoon et al., 2024) and significant divergence from the distribution of real human survey responses (Sun et al., 2024). These limitations may stem from sociocultural biases observed since early models (Brown et al., 2020), creating substantial misalignment with numerous demographic groups (Santurkar et al., 2023; Tao et al., 2024). The "hyper-accuracy distortion" in

human subject study replication (Aher et al., 2023) further demonstrates LLMs' failure to capture diverse viewpoints. Together, these findings highlight fundamental limitations in representing population diversity, raising concerns about LLMs' reliability for statistical inference and their tendency to encode and perpetuate dominant community norms (Bender et al., 2021; Liu et al., 2024). Our work aims at making outputs from LLMs more diverse and representative of a given human population.

**Inducing and aligning behaviors of LLMs**. Recent work has explored techniques to mitigate diversity and representational issues by aligning LLMs with different population groups. LLMs are often treated as a superposition of perspectives (Kovac et al., 2023), which allows different prompting strategies without modifying model parameters, such as in-context impersonation (Salewski et al., 2023) and demographic-aware prompting (Santurkar et al., 2023). Another line of work involves training or fine-tuning models for cross-cultural alignment (Ramezani & Xu, 2023), value alignment (Liu et al., 2022). Our work builds upon the success of leveraging in-context learning and prompting techniques, but differs in a sense that we focus on optimizing prompts for multiple agents to collectively represent a diverse human population, without using demographic information or metadata.

## 3 PROBLEM FORMULATION

In this section, we first introduce the necessary notation and background. We then formally define the problem of selecting a representative set of agents and our optimization objective.

### 3.1 PRELIMINARIES

Let $\mathcal{H} = \{h_1, h_2, \ldots, h_N\}$ denote a population of $N$ humans we want to represent. We assume access to demonstrations (e.g., task–response pairs) from $\mathcal{H}$ on a set of tasks $\mathcal{T}$, denoted by $\mathcal{D}_{\mathcal{H}}^{\mathcal{T}}$. If each human provides a response to every task, then $|\mathcal{D}_{\mathcal{H}}^{\mathcal{T}}| = |\mathcal{H}| \times |\mathcal{T}|$. Each human $h$ is represented by a vector $\mathbf{e}_h \in \mathbb{R}^d$ that summarizes their behavior across the tasks in $\mathcal{T}$. We define an agent $l$ as an LLM conditioned on a set of $K$ demonstrations, where each demonstration is a task–response pair. The demonstrations are used to steer the agent's behavior through in-context learning. The space of all possible agents that can be constructed from $\mathcal{D}_{\mathcal{H}}^{\mathcal{T}}$ is denoted by $|\mathcal{L}| = \binom{|\mathcal{D}_{\mathcal{H}}^{\mathcal{T}}|}{K}$. This number is finite but typically much larger than $|\mathcal{H}|$. When $K$ is fixed, $|\mathcal{L}|$ asymptotically scales as $(|\mathcal{T}| \cdot |\mathcal{H}|)^K$, which we use later for complexity analysis. Similar to humans, each agent $l$ is represented by a vector $\mathbf{e}_l \in \mathbb{R}^d$ that captures its behavior across tasks in $\mathcal{T}$. In our experiments (Section 5), we explore several types of behavioral embeddings, including binary vectors of student performance, vectors of individuals' answer choices, and continuous semantic embeddings of annotators' responses.

We measure the behavioral similarity between any two objects (humans or agents) by the distance between their embedding vectors, defined as $dist(\mathbf{e}_1, \mathbf{e}_2)$ (e.g., Euclidean distance). The *representation gap* of a set of agents $L \subseteq \mathcal{L}$ with respect to the population $\mathcal{H}$ is the average distance between each human $h$ and its closest agent $l \in L$, given by $g(L) = \frac{1}{|\mathcal{H}|} \sum_{h \in \mathcal{H}} \min_{l \in L} dist(\mathbf{e}_h, \mathbf{e}_l)$. This representation gap serves as a proxy for how well the selected agents capture the diverse behaviors and perspectives of the given population. As we will show in our experiments (Section 5), minimizing this gap leads to agents whose behaviors align closely with those of humans on unseen tasks.

### 3.2 OBJECTIVE

Our goal is to select a set of agents $L^{\text{opt}} \subseteq \mathcal{L}$ of size $M$ that best represents the human population by minimizing the average distance between each human and its closest agent: $L^{\text{opt}} = \arg\min_{L \subseteq \mathcal{L}, |L| \leq M} g(L)$. We define the baseline gap when no agents are selected ($L = \emptyset$) as $g(\emptyset) = \frac{1}{|\mathcal{H}|} \sum_{h \in \mathcal{H}} D_{\max}$, where $D_{\max}$ is a constant larger than any possible human–agent embedding distance. It is often more natural to view the problem as maximizing the gain relative to this baseline. We therefore define the average distance reduction obtained by selecting a set $L$ as:

$$f(L) = g(\emptyset) - g(L) = \frac{1}{|\mathcal{H}|} \sum_{h \in \mathcal{H}} \left[ D_{\max} - \min_{l \in L} dist(\mathbf{e}_h, \mathbf{e}_l) \right] \tag{1}$$

This gives us a monotone submodular function $f(L)$ that can be approximated using greedy algorithms. The optimization problem is therefore to find $L^{\text{opt}} = \arg\max_{L \subseteq \mathcal{L}, |L| \leq M} f(L)$.

Table 1: Performance guarantees and time complexity analysis.

| Method | Performance Guarantee | Time Complexity |
|---|---|---|
| OPT | $f(L^{\text{opt}})$ | $\mathcal{O}\big((|\mathcal{T}| \cdot |\mathcal{H}|)^{KM}\big)$ |
| SINGLE | – | $\mathcal{O}(1)$ |
| RANDOM | – | $\mathcal{O}(M)$ |
| GREEDY | $(1 - \frac{1}{e}) \cdot f(L^{\text{opt}})$ | $\mathcal{O}\big(M \cdot |\mathcal{H}| \cdot (|\mathcal{T}| \cdot |\mathcal{H}|)^{K}\big)$ |
| SAMPLEGREEDY | – | $\mathcal{O}\big(M \cdot |\mathcal{H}| \cdot \psi \cdot (|\mathcal{T}| \cdot |\mathcal{H}|)^{K}\big)$ |
| REPPOP$_{\text{demo}}$ (ours) | – | $\mathcal{O}\big(M \cdot K \cdot |\mathcal{T}| \cdot |\mathcal{H}|^2\big)$ |
| REPPOP$_{\text{mapped-1}}$ (ours) | $(1 - 1/e) \cdot (\gamma \cdot f(L^{\text{opt}}) - \rho)$ | $\mathcal{O}\big(K \cdot |\mathcal{H}| + M \cdot |\mathcal{H}|^2\big)$ |
| REPPOP$_{\text{mapped-2}}$ (ours) | $(1 - 1/e) \cdot (\gamma \cdot f(L^{\text{opt}}) - \rho)$ | $\mathcal{O}\big(K \cdot |\mathcal{T}| \cdot |\mathcal{H}| + M \cdot |\mathcal{H}|^2\big)$ |

## 4 METHODOLOGY

In this section, we establish the hardness of the representative agent selection problem and prove the submodularity of our objective function. We then discuss how naive applications of existing strategies are insufficient and present improved methods in Section 4.2. The performance guarantees and time complexity of the methods discussed in this section are summarized in Table 1.

### 4.1 HARDNESS OF THE PROBLEM AND CONNECTION TO SUBMODULARITY

**Theorem 1** (NP-Hardness)**.** *The problem of selecting an optimal subset $L^* \subseteq \mathcal{L}$ of size $M$ that maximizes $f(L)$ is NP-hard.*

*Proof sketch.* We show NP-hardness through a reduction from the uncapacitated facility location problem (UFLP) with zero facility costs, which is known to be NP-hard (Verter, 2011). We provide a detailed proof of this proposition in Appendix D.1.

**Proposition 1** (Submodularity of the Objective Function $f(L)$)**.** *The objective function $f(L) = \frac{1}{|\mathcal{H}|} \sum_{h \in \mathcal{H}} [D_{\max} - \min_{l \in L} \text{dist}(\mathbf{e}_h, \mathbf{e}_l)]$ is submodular.*

*Proof sketch.* Our objective function $f(L)$ exhibits the diminishing returns property of submodularity, which follows directly from its connection to the facility location problem (Krause & Golovin, 2014). We provide proof of this proposition in Appendix D.2.

**Greedy Approximation**. Due to the NP-hardness of the problem, finding the optimal solution $L^{\text{opt}}$ requires exhaustive search with time complexity $\mathcal{O}((|\mathcal{T}| \cdot |\mathcal{H}|)^{KM})$. The submodularity of our objective function enables the GREEDY algorithm to achieve a $(1 - 1/e) \cdot f(L^{\text{opt}})$-approximation guarantee (Nemhauser et al., 1978), with time complexity $\mathcal{O}(M \cdot |\mathcal{H}| \cdot (|\mathcal{T}| \cdot |\mathcal{H}|)^{K})$. However, this approach becomes intractable as the agent space grows. A stochastic variant, STOCHASTICGREEDY, samples a subset of agents at each iteration to approximate the solution (Mirzasoleiman et al., 2015), but this still remains impractical for large spaces. Inspired by (Singla et al., 2014; Mirzasoleiman et al., 2015), we adopt SAMPLEGREEDY, which fixes a candidate pool $\mathcal{C}$ containing only a fraction $\psi$ of agents from $\mathcal{L}$ and applies greedy selection within this pool. At each round, the marginal contribution of each remaining agent in $\mathcal{C}$ is evaluated relative to the current set, and the best agent is added. This process continues until $M$ agents are selected. The time complexity is $\mathcal{O}(M \cdot |\mathcal{H}| \cdot \psi \cdot (|\mathcal{T}| \cdot |\mathcal{H}|)^{K})$, which is still impractical for large populations and motivates the more efficient methods introduced in the next section.

### 4.2 OUR PROPOSED METHODS

**Greedy selection of demonstrations for an agent's context**. Searching through the exponentially large agent space $\mathcal{L}$ is computationally infeasible, limiting the practicality of standard greedy methods. Hence, we propose an alternative method, REPPOP$_{\text{demo}}$ (**Rep**resentative **Pop**ulation using **demo**nstration-level greedy selection), which reduces the complexity to $\mathcal{O}(M \cdot K \cdot |\mathcal{T}| \cdot |\mathcal{H}|^2)$ and empirically achieves competitive performance. Instead of enumerating all candidate agents and evaluating their marginal gains, REPPOP$_{\text{demo}}$ builds each agent incrementally (cf. Algorithm 1). At each step it greedily selects a demonstration from the pool $\mathcal{D}_{\mathcal{H}}^{\mathcal{T}}$ to extend the current context $\Omega$. We denote by $l_{\Omega}$ the agent constructed from $\Omega$. This demonstration-level greedy construction avoids the exponential blow-up in $|\mathcal{L}|$, but sacrifices the formal performance guarantee of standard greedy selection.

---

**Algorithm 1** Greedy selection of demonstrations (REPPOP$_{\text{demo}}$)

---

1: **Input:** Human set $\mathcal{H}$, human demonstrations $\mathcal{D}_{\mathcal{H}}^{\mathcal{T}}$, number of agents to select $M$, context size $K$
2: **Output:** A set of representative agents $L \subseteq \mathcal{L}$ with $|L| \leq M$
3: Initialize $L \leftarrow \emptyset$
4: **for** $i = 1$ to $M$ **do**
5:  Initialize $\Omega \leftarrow \emptyset$                  ▷ Best set of K demonstrations found so far
6:  **for** $k = 1$ to $K$ **do**
7:   $demo^* \leftarrow \arg\max_{demo \in \mathcal{D}_{\mathcal{H}}^{\mathcal{T}} \setminus \Omega} f(L \cup \{l_{\Omega \cup \{demo\}}\}) - f(L)$              ▷ Select demo.
8:   $\Omega \leftarrow \Omega \cup \{demo^*\}$
9:  $L \leftarrow L \cup \{l_\Omega\}$              ▷ Add agent with context $\Omega$ to solution set
10: **return** $L$

---

**Algorithm 2** Greedy selection of human-mapped agents (REPPOP$_{\text{mapped-1}}$ and REPPOP$_{\text{mapped-2}}$)

---

1: **Input:** Human set $\mathcal{H}$, human demonstrations $\mathcal{D}_{\mathcal{H}}^{\mathcal{T}}$, number of agents to select $M$, context size $K$
2: **Output:** A set of representative agents $L \subseteq \mathcal{L}$ with $|L| \leq M$
3: Initialize $\tilde{\mathcal{L}} \leftarrow \emptyset$, $L \leftarrow \emptyset$
4: **for** each human $h \in \mathcal{H}$ **do**
5:  Create agent $l_h$ using a subset of K demonstrations from $\mathcal{D}_h^{\mathcal{T}}$          ▷ Human-mapped agent
6:  $\tilde{\mathcal{L}} \leftarrow \tilde{\mathcal{L}} \cup \{l_h\}$              ▷ Add the agent to pool
7: **for** $i = 1$ to $M$ **do**
8:  $l^* \leftarrow \arg\max_{l \in \tilde{\mathcal{L}} \setminus L} f(L \cup \{l\}) - f(L)$                  ▷ Select agent
9:  $L \leftarrow L \cup \{l^*\}$              ▷ Add agent to solution set
10: **return** $L$

---

**Greedy selection of human-mapped agents.** To further address the computational intractability of searching the full agent space $\mathcal{L}$, we introduce a reduced pool of proxies that directly reflect the humans in the population. We construct $\tilde{\mathcal{L}} = \{l_h \mid h \in \mathcal{H}\}$, where each agent $l_h$ corresponds to a human $h \in \mathcal{H}$ and is formed by conditioning on a subset of $K$ demonstrations from $\mathcal{D}_h^{\mathcal{T}}$. This one-to-one mapping reduces the candidate space to $|\tilde{\mathcal{L}}| = |\mathcal{H}|$ while preserving diversity. The selection problem then becomes $L^* = \arg\max_{L \subseteq \tilde{\mathcal{L}}, |L| \leq M} f(L)$. Building on this human-centered mapping idea, we instantiate two methods, REPPOP$_{\text{mapped-1}}$ and REPPOP$_{\text{mapped-2}}$, which both follow the general procedure in Algorithm 2. Their only difference lies in how demonstrations are selected to construct a human-mapped agent (line 5). In REPPOP$_{\text{mapped-1}}$, the $K$ demonstrations for each human are sampled uniformly at random, yielding lightweight proxy agents with cost $\mathcal{O}(K|\mathcal{H}|)$. In contrast, REPPOP$_{\text{mapped-2}}$ selects the $K$ demonstrations greedily with respect to the human's own behavior, producing stronger proxies at cost $\mathcal{O}(K|\mathcal{T}||\mathcal{H}|)$. Concretely, for each human $h$, the demonstrations are chosen to minimize the distance between the human's embedding $\mathbf{e}_h$ and the embedding of the constructed agent $\mathbf{e}_{l_h}$, i.e., $\text{dist}(\mathbf{e}_h, \mathbf{e}_{l_h})$. Both methods share the same greedy selection stage over the proxy pool, which requires $\mathcal{O}(M|\mathcal{H}|^2)$ time, and both enjoy the same approximation guarantee in Theorem 2.

**Theorem 2** (Performance Guarantee for REPPOP$_{\text{mapped-1}}$ and REPPOP$_{\text{mapped-2}}$). *Let $\tilde{\mathcal{L}} = \{l_h | h \in \mathcal{H}\}$ be the proxy agent set where for each $h \in \mathcal{H}$, $l_h \in N_\rho(h)$, with $N_\rho(h)$ representing the $\rho$-neighborhood of $h$. Define the human coverage ratio $\gamma = f(L_{\mathcal{H}}^*)/f(L_{\mathcal{L}}^*) \in [0, 1]$, where $L_{\mathcal{H}}^*$ is the optimal subset from the human set and $L_{\mathcal{L}}^*$ is the optimal subset from the full agent set. If $L_{\tilde{\mathcal{L}}}^{greedy}$ is the subset of size $M$ returned by the greedy algorithm on $\tilde{\mathcal{L}}$, then:*

$$f(L_{\tilde{\mathcal{L}}}^{greedy}) \geq (1 - 1/e)\left(\gamma \cdot f(L_{\mathcal{L}}^*) - \rho\right),$$

*where $\gamma$ measures the cost of restricting the search space to humans (coverage quality) and $\rho$ measures the cost of approximating each human by a proxy agent (imitation error). The value of $\gamma$ is determined by how expressive the human set is relative to the full agent space, whereas $\rho$ depends on the proxy construction strategy: uniform sampling in REPPOP$_{mapped-1}$ typically yields larger $\rho$, while greedy selection in REPPOP$_{mapped-2}$ achieves smaller $\rho$ at the expense of higher computational cost.*

*Proof sketch.* We show that for the optimal human subset $L_{\mathcal{H}}^*$, the corresponding set of proxy agents $L_{\tilde{\mathcal{H}}}^* = \{l_h | h \in L_{\mathcal{H}}^*\} \subseteq \tilde{\mathcal{L}}$ satisfies $f(L_{\tilde{\mathcal{H}}}^*) \geq f(L_{\mathcal{H}}^*) - \rho$ due to the $\rho$-neighborhood property. Since $L_{\tilde{\mathcal{L}}}^*$ is optimal within $\tilde{\mathcal{L}}$, we have $f(L_{\tilde{\mathcal{L}}}^*) \geq f(L_{\tilde{\mathcal{H}}}^*)$. By the standard greedy approximation for submod-

Table 2: Statistics of datasets used in our experiments.

| Dataset | Domain | Task Type | Repr. Type | Multimodal | No. Humans | No. Tasks | Source |
|---|---|---|---|---|---|---|---|
| EEDI | Education | Multi-choice | Performance | No | 50 | 40 | Primary & High school students |
| OpinionQA | Opinion Survey | Multi-choice | Opinion | No | 500 | 77 | US citizen |
| Wikiart | Image Annotation | Open-ended | Semantic | Yes | 100 | 20 | LLM-based annotators |

A group of 7 friends has 3 chocolate bars that they plan to share equally. 3 new friends arrive and bring along 1 more chocolate bar. There are now 10 friends with 4 chocolate bars to share equally. What has happened to the amount of chocolate each of the original group of 7 now receive?

A. It has increased

B. It has decreased

C. It has stayed the same

D. We need more information

What should be the priority for dealing with illegal immigration in the U.S.?

A. Better border security and stronger enforcement of our immigration

B. Creating a way for immigrants already here illegally to become citizens if they meet certain requirements

C. Both should be given equal priority

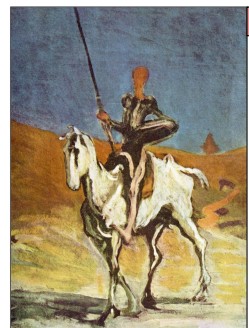

**Emotions:** sadness, pessimism, shyness

**Affective description:**
There's a definite melancholy to this one. The figure on the goat is oddly shaped and indistinct, almost like a shadow, giving a lonely feeling. The muted colors and blurred background really emphasize that sense of isolation. It's not a vibrant scene, but a quiet one, which feels a bit reserved and shy. And overall, it leaves me with a somewhat bleak, pessimistic impression, like something has been lost or is fading away.

(a) Task in EEDI  (b) Task in OpinionQA  (c) Painting in Wikiart with an annotation example.

Figure 2: Examples of tasks in our experiments.

ular functions, $f(L_{\tilde{\mathcal{L}}}^{\text{greedy}}) \geq (1 - 1/e)f(L_{\mathcal{L}}^*)$. Combining these inequalities and using the human coverage ratio $\gamma = f(\tilde{L}_{\mathcal{H}}^*)/f(L_{\mathcal{L}}^*)$, we derive our bound. We provide complete proof in Appendix D.3.

## 5 EXPERIMENTAL EVALUATION

In this section, we present the evaluation domains and datasets (see Table 2 for summary statistics), describe the methods evaluated, and discuss the main results, with additional setup details in Appendix B and further results in Appendix C.

### 5.1 EVALUATION DOMAINS AND DATASETS

**Education: Math Questions and Answers (EEDI).** In this domain, LLMs capturing the behavior of a diverse student population, can allow teachers to practice instructional strategies (Markel et al., 2023) and to conduct virtual pretesting (Benedetto et al., 2024) in a safe environment. Representing students with varying levels of skills and misconceptions can thus benefit both teachers and learners. To this end, we evaluate whether our methods can faithfully represent such a diverse group of students. Specifically, we use the EEDI dataset (Wang et al., 2020), which contains multiple-choice math questions and answers collected from students. Figure 2a shows an example question. We select 50 students and 40 exercises (tasks) from the dataset, splitting tasks and their corresponding answers 50/50 into the training and testing sets. Each student is represented by a binary embedding vector indicating their correct and incorrect answers to the math questions. The representation gap is measured using the L1 distance, which in this case is equivalent to the Hamming distance. Intuitively, it counts the number of questions on which two students' answers differ, reflecting their performance difference.

**Crowdsourcing: Opinion Survey (OpinionQA).** In this application, LLMs can be used as surrogates for crowdworkers, giving answers that reflect different opinions and beliefs that diverse groups of people might express. We use the OpinionQA dataset (Santurkar et al., 2023), which contains multiple-choice questions from the Pew American Trends Panel (ATP) surveys along with human responses. In particular, we focus on the ATP W92 survey, which includes 77 questions related to politics. Figure 2b shows an example question. From this dataset, we sample 500 people and their responses, and split the survey questions into 40 training and 37 test tasks. For each question, answer choices are mapped to ordinal values and normalized to the range $[-1, 1]$. Each human is represented by a vector embedding of their responses, and the representation gap is measured using the L2 distance. This distance captures the extent of differences in opinions and beliefs expressed in their answers.

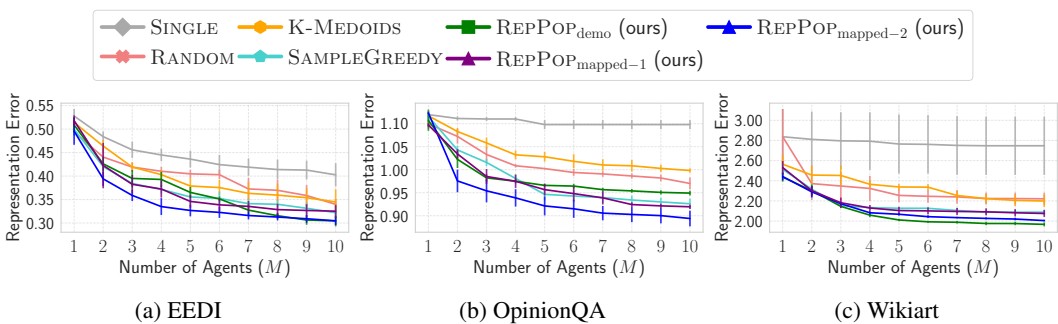

(a) EEDI       (b) OpinionQA       (c) Wikiart

Figure 3: Representation error on test set. We show the representation error on the test set of each method with different number of agents. We report the means and standard errors (error bars) of three runs with different seeds. Our methods maintain lower representation error compared to baselines.

**Crowdsourcing: Data Annotation (WikiArt).** In this application, LLMs can be used as surrogates for crowdworkers for data annotation. We focus on tasks where diverse perspectives are encouraged. For example, to create a datasets on the emotions evoked by art (Mohammad & Kiritchenko, 2018; Mohamed et al., 2022), human crowdworkers were shown a painting and asked to specify the emotions it evoked and to provide a short affective description. Figure 2c shows an example of such a task. Our goal is to construct a set of LLM agents representative of a given pool of annotators in terms of both emotions and language use. We take 20 paintings from the WikiArt dataset (Tan et al., 2019) and split them 50/50 into training and testing tasks. Unfortunately, existing datasets do not provide annotations at the level of individual annotators, and thus we cannot use them directly in our experiments. Hence, we generated 100 "synthetic humans" by prompting LLMs with different "personalities" and asking them to annotate the paintings. Each annotator is then represented by a continuous embedding of their responses extracted from an LLM (cf. Appendix C.3 for details and alternative embedding strategies). Distances between these embeddings are computed using the L2 distance, which captures differences in annotators' perspectives and behaviors across tasks.

## 5.2 METHODS EVALUATED

We compare our proposed methods REPPOP$_{demo}$, REPPOP$_{mapped-1}$, and REPPOP$_{mapped-2}$ from Section 4.2 against SAMPLEGREEDY from Section 4.1 and the following baselines. The SINGLE baseline uniformly samples a single agent from $\mathcal{L}$ and performs $M$ rollouts, while RANDOM baseline uniformly selects $M$ agents from $\mathcal{L}$ and performs one rollout for each. The K-MEDOIDS baseline applies $K$-medoids clustering to form $M$ clusters of humans, and for each cluster uniformly samples $K$ demonstrations from the humans in that cluster to construct an agent; this approach requires re-clustering whenever a new agent is added. For SAMPLEGREEDY, we set the sample size to the number of humans in all experiments. For REPPOP$_{demo}$, we accelerate evaluation with a stochastic greedy variant that samples a subset of $\alpha$ demonstration candidates from the pool $\mathcal{D}_{\mathcal{H}}^{\mathcal{T}}$, setting $\alpha = 100$ for WikiArt and EEDI, and $\alpha = 1000$ for OpinionQA (values chosen based on the scale of $\mathcal{D}_{\mathcal{H}}^{\mathcal{T}}$). All agents are implemented with decoding temperature fixed at 1.0 across all experiments.

## 5.3 RESULTS

**Representation gap**. We compare the considered methods by measuring the representation gap on the test tasks $\mathcal{T}_{\text{test}}$. The error is normalized according to the distance used in each dataset: by $d$ for the EEDI dataset, and by $\sqrt{d}$ for the OpinionQA and WikiArt datasets. Figure 3 shows the normalized test representation error of agent sets constructed by each method using Gemma3-12B as the underlying LLM, with varying numbers of agents (training plots are provided in Appendix C). Using a randomly sampled set of $N$ agents (RANDOM) reduces the representation error compared to using a single agent with $N$ rollouts (SINGLE), emphasizing the importance of the problem addressed in this work. The heuristic method K-MEDOIDS does not improve over the simple baseline RANDOM. By exploiting the submodularity of the problem, SAMPLEGREEDY applies a greedy selection procedure on a sampled subset of agents and achieves lower representation error than the aforementioned baselines. Finally, our methods further reduce representation error across all datasets.

Table 3: Generalization to other models. Representation error on the EEDI dataset with $M = 10$ and $K = 3$ across model families of varying sizes (4B–70B). Results are reported on the test set, with bold numbers indicating the lowest error. Our proposed methods consistently outperform baselines, demonstrating lower representation error across all tested models and highlighting the robustness of our framework independent of the underlying model choice.

| Method | Model | | | | | |
|---|---|---|---|---|---|---|
| | Phi-4-mini (4B) | Phi-4 (14B) | Qwen-3 (14B) | Gemma-3 (27B) | Qwen-3 (32B) | Llama-3.1 (70B) |
| SINGLE | 0.39 | 0.39 | 0.42 | 0.35 | 0.37 | 0.46 |
| RANDOM | 0.39 | 0.36 | 0.36 | 0.35 | 0.35 | 0.46 |
| K-MEDOIDS | 0.41 | 0.36 | 0.38 | 0.36 | 0.37 | 0.38 |
| SAMPLEGREEDY | 0.38 | 0.36 | 0.34 | 0.34 | 0.34 | 0.37 |
| REPPOP$_{demo}$ | 0.37 | **0.30** | 0.33 | **0.31** | **0.33** | **0.35** |
| REPPOP$_{mapped-1}$ | 0.38 | 0.35 | 0.34 | 0.32 | 0.35 | 0.38 |
| REPPOP$_{mapped-2}$ | **0.35** | 0.33 | **0.30** | 0.32 | 0.35 | 0.38 |

Table 4: Average runtime (in minutes) for selecting an agent, reported as mean $\pm$ standard error over three seeds. We use agent set size $M = 10$, context size $K = 3$, and the Gemma3-12B model.

| Method | Dataset | | | | | | | | |
|---|---|---|---|---|---|---|---|---|---|
| | EEDI | | | OpinionQA | | | Wikiart | | |
| | $K=1$ | $K=3$ | $K=5$ | $K=1$ | $K=3$ | $K=5$ | $K=1$ | $K=3$ | $K=5$ |
| SINGLE | $0.3 \pm 0.0$ | $0.3 \pm 0.0$ | $0.3 \pm 0.0$ | $0.6 \pm 0.0$ | $0.4 \pm 0.0$ | $0.4 \pm 0.0$ | $2.0 \pm 1.0$ | $1.8 \pm 0.9$ | $1.6 \pm 0.6$ |
| RANDOM | $0.3 \pm 0.0$ | $0.3 \pm 0.0$ | $0.3 \pm 0.0$ | $0.3 \pm 0.0$ | $0.3 \pm 0.0$ | $0.3 \pm 0.0$ | $3.2 \pm 1.8$ | $1.8 \pm 0.8$ | $1.6 \pm 0.6$ |
| K-MEDOIDS | $0.9 \pm 0.0$ | $1.1 \pm 0.0$ | $1.3 \pm 0.0$ | $1.6 \pm 0.0$ | $1.7 \pm 0.0$ | $0.9 \pm 0.0$ | $2.5 \pm 0.0$ | $1.9 \pm 0.0$ | $2.3 \pm 0.0$ |
| SAMPLEGREEDY | $0.6 \pm 0.0$ | $0.7 \pm 0.0$ | $0.8 \pm 0.0$ | $2.7 \pm 0.1$ | $2.8 \pm 0.0$ | $3.2 \pm 0.0$ | $2.2 \pm 0.0$ | $1.8 \pm 0.0$ | $2.2 \pm 0.0$ |
| REPPOP$_{demo}$ | $6.4 \pm 0.1$ | $21.3 \pm 0.2$ | $40.1 \pm 0.4$ | $38.3 \pm 0.3$ | $108.4 \pm 0.9$ | $179.0 \pm 31.7$ | $16.4 \pm 0.1$ | $43.6 \pm 0.3$ | $78.2 \pm 0.2$ |
| REPPOP$_{mapped-1}$ | $0.6 \pm 0.0$ | $0.7 \pm 0.0$ | $0.8 \pm 0.0$ | $2.7 \pm 0.1$ | $2.9 \pm 0.0$ | $3.2 \pm 0.1$ | $2.2 \pm 0.0$ | $1.8 \pm 0.0$ | $2.2 \pm 0.0$ |
| REPPOP$_{mapped-2}$ | $5.9 \pm 0.0$ | $17.8 \pm 0.0$ | $30.9 \pm 0.1$ | $71.7 \pm 0.3$ | $197.1 \pm 0.3$ | $350.1 \pm 0.9$ | $20.2 \pm 0.2$ | $43.0 \pm 0.1$ | $66.8 \pm 0.5$ |

Notably, REPPOP$_{mapped-2}$ outperforms the strongest baseline SAMPLEGREEDY significantly on all three datasets, with $p < 0.01$ according to a paired t-test where pairs are formed by matching runs that share the same random seed and the same number of agents. These results demonstrate that our methods can construct agent sets that represent human population behavior and generalize effectively.

**Generalization to other LLMs.** We conduct experiments with a variety of models to examine how method performance changes across model families and sizes. Due to resource constraints, this analysis is limited to the EEDI dataset with $M = 10$ and $K = 3$. Table 3 shows that the same trends hold across different LLMs in terms of reducing representation error. Our proposed methods consistently outperform the baselines, achieving lower representation error on the test set across all tested models. These results highlight the robustness of our framework regardless of the underlying LLM.

**Trade-offs between performance and computation.** We investigate the trade-off between performance and computational cost of different methods. Table 4 reports the average runtime for selecting an agent (details of computational resources are provided in Appendix B.4). Our method REPPOP$_{mapped-1}$ runs as fast as SAMPLEGREEDY while achieving slightly better performance. Both REPPOP$_{demo}$ and REPPOP$_{mapped-1}$ are more computationally expensive but offer more representative agent sets. In practice, these trade-offs should be considered when selecting a suitable method.

### 5.4 AGENT BEHAVIOR ANALYSIS

Beyond measuring the representation gap via embedding distances, we investigate whether the agents exhibit behaviors similar to the humans they represent in the EEDI and OpinionQA datasets (cf. Appendix C.3 for analysis on WikiArt). First, we analyze, for each agent, the behavior of the humans it represents. Figure 4 shows 2D embeddings of humans and agents constructed by our method REPPOP$_{mapped-2}$, computed on the training set. We observe that REPPOP$_{mapped-2}$ constructs agents that cover different regions of the human embedding space, corresponding to distinct groups of humans. Next, we analyze whether these agents behave like the groups of people they are intended to represent.

**EEDI dataset.** We select three agents and visualize the students they represent, along with the aggregated math skill levels of those students in Figure 4a. For example, Agent 4 represents students with a strong understanding of Mental Multiplication and Division but weaker proficiency in Fractions and Negative Numbers. We then use these agents as surrogates for students to answer

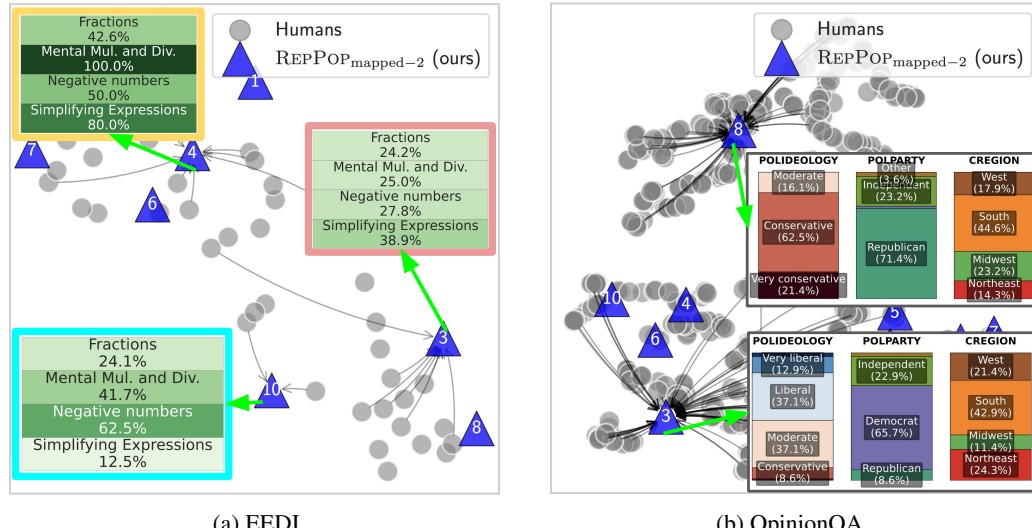

(a) EEDI  (b) OpinionQA

Figure 4: 2D embeddings of humans and agents constructed by REPPOP$_{\text{mapped-2}}$ on tasks in $\mathcal{T}_{\text{train}}$ using UMAP. We provide examples of aggregated metadata (in the boxes) of humans represented by agents (connections are denoted by black arrows). They are not used for constructing agents and used only for analysis. Our method REPPOP$_{\text{mapped-2}}$ constructs agents to cover different human behaviors, collectively (approximately) representing the human population. **(a) EEDI:**. Each agent represents a group of students with particular success rates on different Math concepts. **(b) OpinionQA:** Each agent represents a group of people with particular distributions of political ideologies, parties, and regions.

new math questions in the test data. The distinct proficiency levels of agents $(3, 4, 10)$ on these new questions are shown in Figure 1, using the same color-coding of boxes. We find that their proficiency levels across different math concepts closely mirror those observed in the students they represent.

**OpinionQA dataset.** We select two agents and visualize the aggregated self-declared metadata of the humans they represent in Figure 4b. For each task in the test set, we map answer choices to political ideologies and party affiliations. For each agent, we compute the distribution over these labels across all questions. For example, for Agent 3 we obtain the following distributions: political ideologies — Very liberal (13.3%), Liberal (40%), Moderate (13.3%), Conservative (26.7%), Very conservative (6.2%); political parties — Independent (13.3%), Democrat (53.3%), Republican (33.3%). These distributions mirror the actual human demographic patterns for Agent 3 shown in Figure 4b, despite the fact that such metadata was not used in constructing the agents. These findings highlight their ability to capture and reproduce meaningful population-level diversity.

## 6 CONCLUDING DISCUSSIONS

We studied the problem of constructing a set of generative agents that collectively represent a given human population. Through formulating it as a submodular optimization problem, we proposed methods that allow different tradeoffs of computational complexity and performance (guarantees). In addition, we empirically demonstrated that our methods can construct a set of representative agents for different populations of annotators and students in crowdsourcing and educational settings. We also demonstrated quantitatively and qualitatively the generalizability of our methods to unseen tasks. Next, we would like to discuss the limitations of our work and propose directions for future research to tackle them. First, we rely on prompting-based approaches rather than fine-tuning; future work could explore fine-tuning techniques for constructing an agent. Second, we do not consider the ordering of demonstrations in prompts, and understanding how ordering influences model behavior would be a valuable extension. Third, while our results show that agents constructed by our methods effectively represent a human population, this work primarily provides groundwork for more comprehensive evaluations of downstream applications, such as using the agents for training teachers, assessing the effectiveness of interventions, or simulating responses to government policy.

ETHICS STATEMENT

This work uses publicly available datasets (EEDI, OpinionQA, and WikiArt) that have been anonymized and do not contain personally identifiable information. No new human subjects were recruited, and thus no IRB approval was required. We respect privacy by using demographic information only for analysis. Potential risks of this work include misuse of representative agents to simulate or stereotype demographic groups. Our methods are intended solely for research and educational purposes and not for applications that could cause harm to individuals or communities. We acknowledge that language models can reflect and amplify biases present in their training data, and our work should not be interpreted as providing perfect or unbiased representations of populations. During preparing this submission, we used Large Language Models (LLMs) for suggesting minor edits, such as grammar, wording, and typos.

REPRODUCIBILITY STATEMENT

We have taken several steps to facilitate the reproducibility of our work. All proposed algorithms and implementation details are provided in the main text and appendices, as well as in the anonymized source code repository. Detailed dataset descriptions, data processing steps, hyperparameters, configurations, and evaluation procedures are documented in Appendix B. We report main results across three random seeds (1, 2, 3) and include variance estimates where appropriate. Complete proofs of our theoretical claims are presented in Appendix D.

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

# APPENDIX

## A    TABLE OF CONTENTS

In this section, we briefly describe the content provided in the paper's appendices.

- Section B provides more details about the experimental setup, including datasets and prompts used for each domain, computational resources.
- Section C provides results on multiple runs with different context sizes $K$ and random seeds.
- Section D provides detailed proofs of our theorems and propositions.

## B   ADDITIONAL EXPERIMENTAL SETUP

### B.1   EEDI

**Dataset.** We use the EEDI math dataset (Wang et al., 2020), which provides data on students' answers to mathematics questions on the Eedi platform. The questions are multiple-choice with four answer choices presented as images, and we select 40 that can be converted to text and have a large number of student responses, chosen to maximize both the number of questions answered per student and the number of students answering each question. The selected questions cover four concepts—Fractions, Negative Numbers, Mental Multiplication and Division, and Simplifying Expressions—with half of the questions in each concept used for training and the other half for testing.

**Prompt for agent.** Each agent is given a set of $K$ demonstrations, which are pairs of math questions and example answers provided by the real students (cf. Figure 5). Then, we ask the agent to analyze the given answers and predict how the student would answer a new question.

---

**Figure 5: [EEDI] Prompt for Agent with K demonstrations**

**[User message]**
{question_1}
{example_answer_of_question_1}
.
.
.
{question_K}
{example_answer_of_question_K}
Evaluate whether the student's previous answers reveal any misconceptions. If so, analyze those misconceptions before proceeding. If not, directly predict how the student would answer the following question.
{question}

---

### B.2   OPINIONQA

**Dataset.** We use questions and answers from the US citizens in the American Trends Panel W92 survey data, which was used in (Santurkar et al., 2023). This survey includes 77 multiple-choice questions related to politics and responses from over $10,000$ respondents across the US. The answer choices typically have an ordinal structure (e.g., ranging from "A great deal" to "Not at all") and we reuse the mapping from answer choices to ordinal values from (Santurkar et al., 2023). In addition, the survey data includes demographic information of the people, including ideology, political party and region. We sample $N = 500$ people and take their answers to create our dataset. We use the demographic information of these people for analysis purposes only.

**Prompt for agent.** Each agent is given a set of $K$ demonstrations, which are pairs of questions and example answers provided by the real survey respondents (cf. Figure 6). Then, we ask the agent to act as the human who has given the answers above and to answer a new question.

---

**Figure 6: [OpinionQA] Prompt for Agent with K demonstrations**

**[User message]**
{question_1}
{example_answer_of_question_1}
.
.
.
{question_K}
{example_answer_of_question_K}
Act as the human who has given the answers above. Answer the following question.
{question}

---

## B.3 WIKIART

---

**Figure 7: [Wikiart] Prompt for Synthetic Human**

**[System message]** Act as a human who has taken the following personality test. Be mindful of how you focus your attention and the way you express yourself through language.
I see myself as someone who is generally trusting: {answer}
I see myself as someone who tends to be lazy: {answer}
I see myself as someone who is relaxed, handles stress well: {answer}
I see myself as someone who has few artistic interests: {answer}
I see myself as someone who is outgoing, sociable: {answer}
I see myself as someone who tends to find fault with others: {answer}
I see myself as someone who does a thorough job: {answer}
I see myself as someone who gets nervous easily: {answer}
I see myself as someone who has an active imagination: {answer}
**[User message]** What emotions does the following painting evoke? Choose from the following list of emotions: gratitude, happiness, humility, love, optimism, trust, anger, arrogance, disgust, fear, pessimism, regret, sadness, shame, agreeableness, anticipation, disagreeableness, shyness, surprise. Provide a short explanation referencing specific details from the painting. Respond in JSON format with two keys: "emotions" and "explanation".
{painting}

---

**Figure 8: [Wikiart] Prompt for extracting embeddings for an annotator**

**[User message]**
Response_1: annotation_of_painting_1
.
.
.
Response_q: annotation_of_painting_q

---

**Figure 9: [Wikiart] Prompt for Agent with K demonstrations**

**[User message]**
{painting_1}
{annotation_of_painting_1}
.
.
.
{painting_K}
{annotation_of_painting_K}
Act as the human who has given the answers above. What emotions does the following painting evoke? Choose from the following list of emotions: gratitude, happiness, humility, love, optimism, trust, anger, arrogance, disgust, fear, pessimism, regret, sadness, shame, agreeableness, anticipation, disagreeableness, shyness, surprise. Provide a short explanation referencing specific details from the painting. Respond in JSON format with two keys: "emotions" and "explanation".
{painting}

---

**Dataset.** We use LLMs conditioned on the Big Five personality traits (McCrae & John, 1992), namely Extraversion, Agreeableness, Conscientiousness, Emotional Stability, and Openness, to act as annotators. Each annotator is specified by responses to a 10-item BFI questionnaire (Rammstedt & John, 2007), which we place in the system message to instruct the LLM to act as a human who has taken the test (cf. Figure 7). Responses take one of five values (Disagree Strongly, Disagree a Little, Neither Agree nor Disagree, Agree a Little, Agree Strongly), which we convert into trait scores using the formula in (Rammstedt & John, 2007) and rescale to $[0, 5]$, where higher values

indicate stronger expression of the trait. To create a diverse set of annotators, we sample responses from normal distributions of trait values with varying means and standard deviations. These system messages condition the LLM to act with distinct personalities, and we use Gemma3-27B (Kamath et al., 2025) to generate their answers.

To create an embedding for each human or agent, we concatenate their answers on the train/test tasks into a single prompt (cf. Figure 8), pass it through a language model (Gemma3-12B (Kamath et al., 2025)), and extract the last hidden state with mean pooling to obtain the embedding. We then reduce the dimensionality to 64 using PCA and measure distances between embeddings using L2 distance.

**Prompt for agent.** Each agent is given a set of $K$ demonstrations, which are pairs of paintings and example annotations provided by the annotators (cf. Figure 9). Then, we ask the agent to act as the annotator who has given the answers above and to annotate a new painting. The LLM agent is given a list of emotions (from (Mohammad & Kiritchenko, 2018)) to choose from, and it must provide a short explanation referencing specific details from the painting.

### B.4    RESOURCES

We use a machine with 2 x Intel Xeon Gold 5317 for all experiments. We use 1 x NVIDIA H100 80GB for experiments on EEDI and OpinionQA datasets, and 1 x NVIDIA H200 141GB for Wikiart dataset.

## C    ADDITIONAL EXPERIMENTAL RESULTS

In this section, we show results on different context sizes $K = 1, 3, 5$ for each dataset. We report the mean and standard error (shown as error bars) computed over three seeds. Overall, we observe similar trends where our methods outperform other baselines across all datasets.

### C.1    EEDI

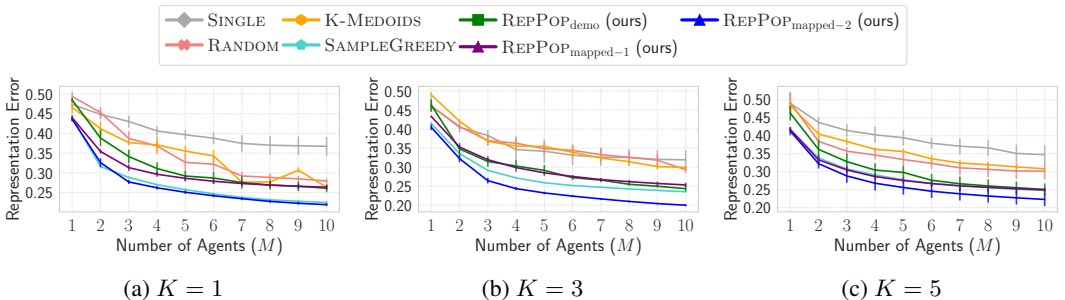

(a) $K = 1$            (b) $K = 3$            (c) $K = 5$

Figure 10: Representation error on EEDI dataset (Train).

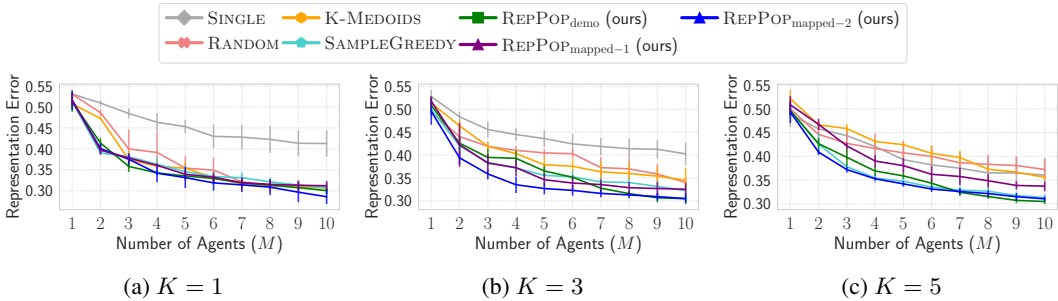

(a) $K = 1$            (b) $K = 3$            (c) $K = 5$

Figure 11: Representation error on EEDI dataset (Test).

## C.2 OPINIONQA

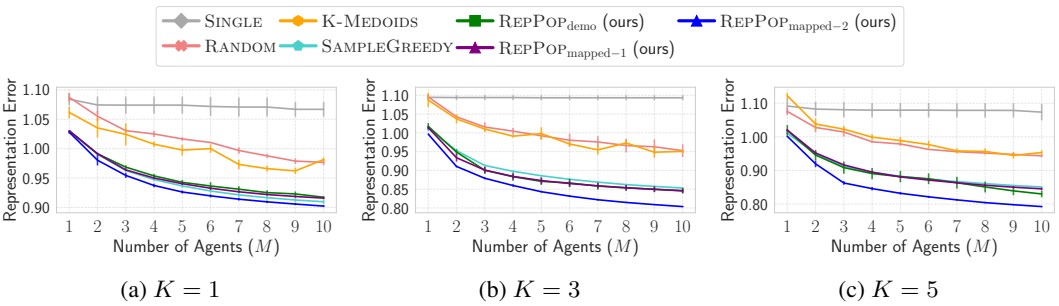

(a) $K = 1$ (b) $K = 3$ (c) $K = 5$

Figure 12: Representation error on OpinionQA dataset (Train).

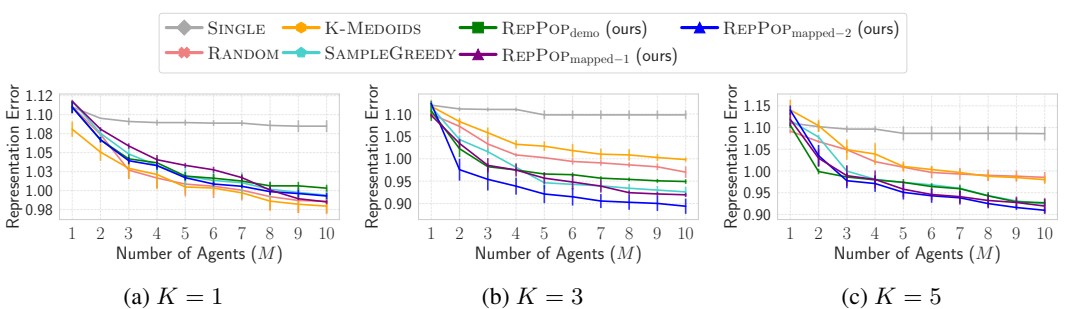

(a) $K = 1$ (b) $K = 3$ (c) $K = 5$

Figure 13: Representation error on OpinionQA dataset (Test).

## C.3 WIKIART

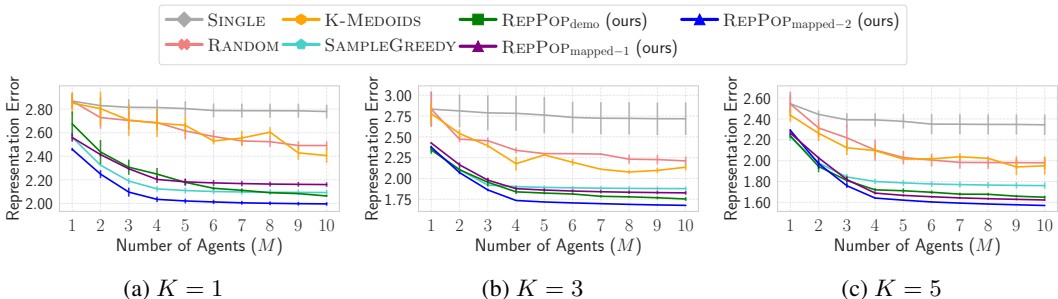

Figure 14: Representation error on Wikiart dataset (Train).

(a) $K = 1$      (b) $K = 3$      (c) $K = 5$

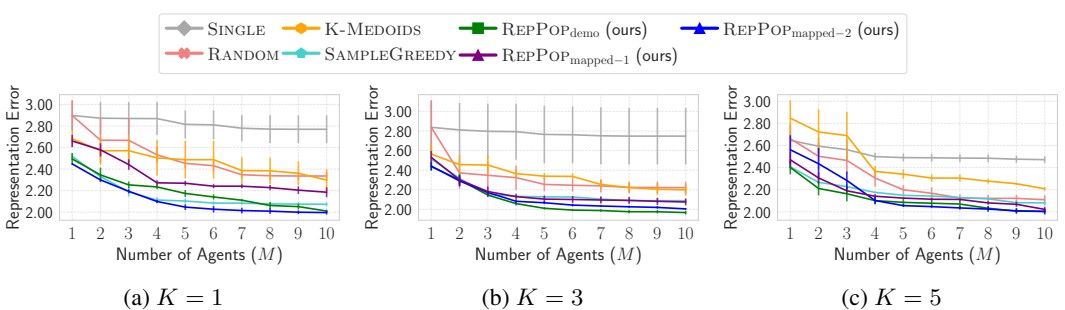

(a) $K = 1$      (b) $K = 3$      (c) $K = 5$

Figure 15: Representation error on Wikiart dataset (Test).

**Agent behavior analysis.** We select three agents and visualize the annotators they represent along with their aggregated traits (cf. Figure 16). For example, Agent 2 represents annotators with high neuroticism and low conscientiousness. To validate whether the agents exhibit behaviors consistent with the annotators they represent, we ask each constructed agent to complete the 10-item BFI test (Rammstedt & John, 2007), and their responses reveal traits that align with the average traits of the corresponding humans. For instance, Agent 2 in Figure 16 obtains the following trait scores: Extraversion (1.0), Agreeableness (1.5), Conscientiousness (1.38), Neuroticism (4.66), and Openness (2.5).

**Embedding model analysis.** We further evaluate our approach on the Wikiart dataset using a smaller and more efficient bidirectional encoder (gte-base-en-v1.5). The results in Table 5 confirm that our methods remain effective even when relying on a much smaller embedding model.

Table 5: Comparison of methods on the Wikiart dataset using the gte-base-en-v1.5 embedding model. We report the representation error on the test set with context size $K = 3$ and agent set size $M = 10$.

| Method | gte-base-en-v1.5 (137M) |
|---|---|
| SINGLE | 0.71 |
| RANDOM | 0.69 |
| K-MEDOIDS | 0.68 |
| SAMPLEGREEDY | 0.68 |
| REPPOP$_{demo}$ (ours) | **0.62** |
| REPPOP$_{mapped-1}$ (ours) | 0.66 |
| REPPOP$_{mapped-2}$ (ours) | 0.68 |

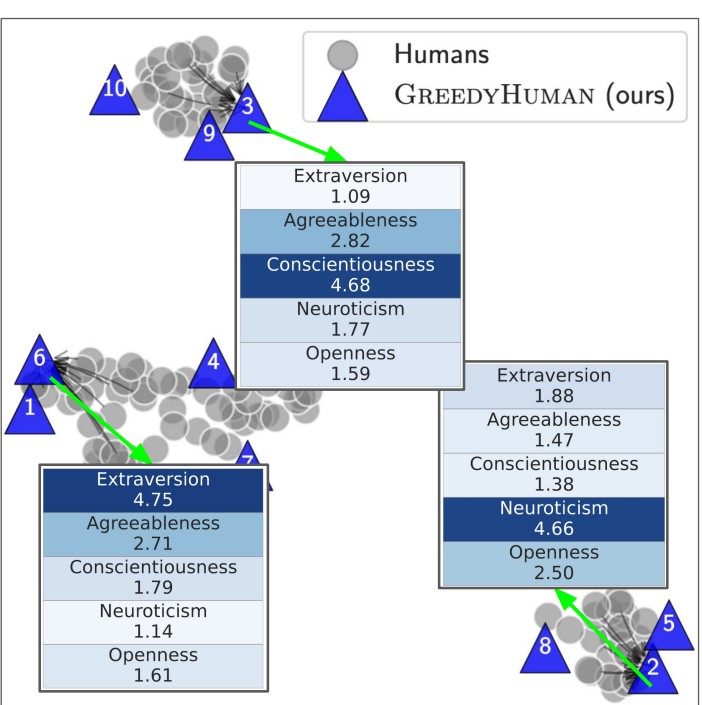

Figure 16: 2D embeddings of humans and agents constructed by REPPOP$_{\text{mapped-2}}$ on $\mathcal{T}_{\text{train}}$ tasks in Wikiart dataset using UMAP (McInnes & Healy, 2018). We provide examples of aggregated traits (heatmaps) of annotators represented by each agent (connections are denoted by black arrows). We note that metadata are not used for constructing agents and used only for analysis. Our method REPPOP$_{\text{mapped-2}}$ constructs agents to cover different areas in the annotator embedding space, collectively representing the annotators. Each agent represents a group of annotators with particular personality traits.

# D PROOFS

## D.1 PROOF OF THEOREM 1

**Theorem 1** (NP-Hardness). The problem of selecting an optimal subset $L^* \subseteq \mathcal{L}$ of size $M$ that maximizes $f(L)$ is NP-hard.

*Proof.* We show NP-hardness through a reduction from the k-facility location problem which extends the uncapacitated facility location problem (UFLP) by including a constraint on the maximum number of facilities. The problem is known to be NP-hard. (Fowler et al., 1981)

Consider an instance of the k-facility location problem with a set of potential facility locations $\mathcal{F} = \{f_1, f_2, \ldots, f_{n_F}\}$, a set of customers $\mathcal{C} = \{c_1, c_2, \ldots, c_{n_C}\}$, service costs $d(f, c)$ representing the cost of serving customer $c$ from facility $f$, and a cardinality constraint $M$. The objective is to select a subset $F \subseteq \mathcal{F}$ of $M$ facilities to minimize the sum of service costs $\sum_{c \in \mathcal{C}} \min_{f \in F} d(f, c)$.

We construct a corresponding instance of our representative agent selection problem as follows: set $\mathcal{H} = \mathcal{C}$ (each customer corresponds to a human); set $\mathcal{L} = \mathcal{F}$ (each potential facility location corresponds to a potential agent); define $\mathrm{dist}(\mathbf{e}_h, \mathbf{e}_l) = d(l, h)$ for each human $h \in \mathcal{H}$ and agent $l \in \mathcal{L}$; and set $M$ to be the number of facilities we wish to open.

Under this construction, our objective function becomes:

$$f(L) = \frac{1}{|\mathcal{H}|} \sum_{h \in \mathcal{H}} \left[ D_{\max} - \min_{l \in L} d(l, h) \right]$$

$$= \frac{1}{|\mathcal{H}|} \left( \sum_{h \in \mathcal{H}} D_{\max} \right) - \frac{1}{|\mathcal{H}|} \sum_{h \in \mathcal{H}} \min_{l \in L} d(l, h)$$

$$= D_{\max} - \frac{1}{|\mathcal{H}|} \sum_{h \in \mathcal{H}} \min_{l \in L} d(l, h)$$

Since $D_{\max}$ is a constant and $\frac{1}{|\mathcal{H}|}$ is a positive constant, maximizing $f(L)$ is equivalent to minimizing $\sum_{h \in \mathcal{H}} \min_{l \in L} d(l, h)$, which is the objective of the k-facility location problem.

Therefore, if the representative agent selection problem could be solved in polynomial time, the k-facility location problem could also be solved in polynomial time. Since the k-facility location problem is NP-hard, the representative agent selection problem must also be NP-hard. □

## D.2 PROOF OF PROPOSITION 1

**Proposition 1** (Submodularity). The objective function

$$f(L) = \frac{1}{|\mathcal{H}|} \sum_{h \in \mathcal{H}} \left[ D_{\max} - \min_{l \in L} \mathrm{dist}(\mathbf{e}_h, \mathbf{e}_l) \right]$$

is submodular.

*Proof.* A set function $f$ is submodular if for all $A \subseteq B \subseteq \mathcal{L}$ and for all $l' \in \mathcal{L} \setminus B$, we have $f(A \cup \{l'\}) - f(A) \geq f(B \cup \{l'\}) - f(B)$.

Let $A \subseteq B \subseteq \mathcal{L}$ and $l' \in \mathcal{L} \setminus B$. The marginal gain of adding $l'$ to $A$ is:

$$f(A \cup \{l'\}) - f(A) = \frac{1}{|\mathcal{H}|} \sum_{h \in \mathcal{H}} \left[ D_{\max} - \min_{l \in A \cup \{l'\}} \mathrm{dist}(\mathbf{e}_h, \mathbf{e}_l) - \left( D_{\max} - \min_{l \in A} \mathrm{dist}(\mathbf{e}_h, \mathbf{e}_l) \right) \right]$$

This simplifies to:

$$f(A \cup \{l'\}) - f(A) = \frac{1}{|\mathcal{H}|} \sum_{h \in \mathcal{H}} \left[ \min_{l \in A} \mathrm{dist}(\mathbf{e}_h, \mathbf{e}_l) - \min_{l \in A \cup \{l'\}} \mathrm{dist}(\mathbf{e}_h, \mathbf{e}_l) \right]$$

Define $\mathcal{H}_A^{l'} = \{h \in \mathcal{H} \mid \text{dist}(\mathbf{e}_h, \mathbf{e}_{l'}) < \min_{l \in A} \text{dist}(\mathbf{e}_h, \mathbf{e}_l)\}$ as the set of humans for whom $l'$ provides improvement when added to $A$. Similarly define $\mathcal{H}_B^{l'}$ for set $B$.

For humans in $h \in \mathcal{H} \setminus \mathcal{H}_A^{l'}$, the closest agent in $A$ remains closer than $l'$, so $\min_{l \in A \cup \{l'\}} \text{dist}(\mathbf{e}_h, \mathbf{e}_l) = \min_{l \in A} \text{dist}(\mathbf{e}_h, \mathbf{e}_l)$, giving zero marginal improvement. Therefore:

$$f(A \cup \{l'\}) - f(A) = \frac{1}{|\mathcal{H}|} \sum_{h \in \mathcal{H}_A^{l'}} \left[ \min_{l \in A} \text{dist}(\mathbf{e}_h, \mathbf{e}_l) - \text{dist}(\mathbf{e}_h, \mathbf{e}_{l'}) \right]$$

Since $A \subseteq B$, for any human $h$, we have $\min_{l \in A} \text{dist}(\mathbf{e}_h, \mathbf{e}_l) \geq \min_{l \in B} \text{dist}(\mathbf{e}_h, \mathbf{e}_l)$.

This implies that if $l'$ provides improvement over $B$ (i.e., $h \in \mathcal{H}_B^{l'}$), then $l'$ also provides improvement over $A$ (i.e., $h \in \mathcal{H}_A^{l'}$). Therefore, $\mathcal{H}_B^{l'} \subseteq \mathcal{H}_A^{l'}$.

For humans in $\mathcal{H}_B^{l'}$, the improvement when adding $l'$ to $A$ is at least as large as when adding $l'$ to $B$, since $\min_{l \in A} \text{dist}(\mathbf{e}_h, \mathbf{e}_l) - \text{dist}(\mathbf{e}_h, \mathbf{e}_{l'}) \geq \min_{l \in B} \text{dist}(\mathbf{e}_h, \mathbf{e}_l) - \text{dist}(\mathbf{e}_h, \mathbf{e}_{l'})$.

Similarly, for $B$ we have:

$$f(B \cup \{l'\}) - f(B) = \frac{1}{|\mathcal{H}|} \sum_{h \in \mathcal{H}_B^{l'}} \left[ \min_{l \in B} \text{dist}(\mathbf{e}_h, \mathbf{e}_l) - \text{dist}(\mathbf{e}_h, \mathbf{e}_{l'}) \right]$$

Therefore:

$$\begin{aligned}
f(A \cup \{l'\}) - f(A) &= \frac{1}{|\mathcal{H}|} \sum_{h \in \mathcal{H}_A^{l'}} \left[ \min_{l \in A} \text{dist}(\mathbf{e}_h, \mathbf{e}_l) - \text{dist}(\mathbf{e}_h, \mathbf{e}_{l'}) \right] \\
&= \frac{1}{|\mathcal{H}|} \sum_{h \in \mathcal{H}_B^{l'}} \left[ \min_{l \in A} \text{dist}(\mathbf{e}_h, \mathbf{e}_l) - \text{dist}(\mathbf{e}_h, \mathbf{e}_{l'}) \right] \\
&\quad + \frac{1}{|\mathcal{H}|} \sum_{h \in \mathcal{H}_A^{l'} \setminus \mathcal{H}_B^{l'}} \left[ \min_{l \in A} \text{dist}(\mathbf{e}_h, \mathbf{e}_l) - \text{dist}(\mathbf{e}_h, \mathbf{e}_{l'}) \right] \\
&\geq \frac{1}{|\mathcal{H}|} \sum_{h \in \mathcal{H}_B^{l'}} \left[ \min_{l \in B} \text{dist}(\mathbf{e}_h, \mathbf{e}_l) - \text{dist}(\mathbf{e}_h, \mathbf{e}_{l'}) \right] \\
&= f(B \cup \{l'\}) - f(B)
\end{aligned}$$

This shows that $f$ is submodular. $\qquad\square$

### D.3 PROOF OF THEOREM 2

**Theorem 2** (Performance Guarantee for REPPOP$_{\text{mapped-1}}$ and REPPOP$_{\text{mapped-2}}$). Let $\tilde{\mathcal{L}} = \{l_h \mid h \in \mathcal{H}\}$ be the proxy agent set where for each $h \in \mathcal{H}$, $l_h \in N_\rho(h)$, with $N_\rho(h)$ representing the $\rho$-neighborhood of $h$. Define the human coverage ratio $\gamma = \frac{f(L_\mathcal{H}^*)}{f(L_\mathcal{L}^*)} \in [0, 1]$, where $L_\mathcal{H}^*$ is the optimal subset from the human set and $L_\mathcal{L}^*$ is the optimal subset from the full agent set. If $L_{\tilde{\mathcal{L}}}^{\text{greedy}}$ is the subset of size $M$ returned by the greedy algorithm on $\tilde{\mathcal{L}}$, then:

$$f(L_{\tilde{\mathcal{L}}}^{greedy}) \geq (1 - \frac{1}{e})(\gamma \cdot f(L_\mathcal{L}^*) - \rho)$$

*Proof.* In our context, $L_\mathcal{H}^* \subseteq \mathcal{H}$ represents the optimal subset of humans of size $M$ that would be selected if we directly choose humans instead of agents. This is a theoretical construct for analysis purposes. In contrast, $L_\mathcal{L}^*$ is the optimal subset of size $M$ from the actual agent set $\mathcal{L}$. The human coverage ratio $\gamma = \frac{f(L_\mathcal{H}^*)}{f(L_\mathcal{L}^*)} \in [0, 1]$ measures how well selecting from the human set can approximate

the optimal solution achievable using the full agent set. We start our proof by first decomposing the objective function as

$$f(L) = \sum_{h \in \mathcal{H}} f_h(L)$$

where

$$f_h(L) = \frac{1}{|\mathcal{H}|} \left[ D_{\max} - \min_{l \in L} \text{dist}(\mathbf{e}_h, \mathbf{e}_l) \right]$$

For each human $h$ in the optimal subset $L_{\mathcal{H}}^*$, consider its corresponding agent $l_h$ in the proxy agent set $\tilde{\mathcal{L}}$. Let $L_{\tilde{\mathcal{H}}}^* = \{l_h | h \in L_{\mathcal{H}}^*\}$.

By definition, for each human $h \in \mathcal{H}$ and its corresponding proxy agent $l_h \in \tilde{\mathcal{L}}$, we have $\text{dist}(\mathbf{e}_h, \mathbf{e}_{l_h}) \leq \rho$ since $l_h$ is in the $\rho$-neighborhood of $h$. Therefore:

$$|f_h(L_{\mathcal{H}}^*) - f_h(L_{\tilde{\mathcal{H}}}^*)| \leq \frac{\rho}{|\mathcal{H}|}$$

The above inequality holds because the maximum distance deviation between any human and its proxy agent is at most $\rho$ (by definition of the $\rho$-neighborhood).

Then:

$$|f(L_{\mathcal{H}}^*) - f(L_{\tilde{\mathcal{H}}}^*)| = \left| \sum_{h \in \mathcal{H}} f_h(L_{\mathcal{H}}^*) - f_h(L_{\tilde{\mathcal{H}}}^*) \right| \leq \sum_{h \in \mathcal{H}} |f_h(L_{\mathcal{H}}^*) - f_h(L_{\tilde{\mathcal{H}}}^*)| \leq \frac{\rho}{|\mathcal{H}|} \cdot |\mathcal{H}| = \rho$$

This gives us:

$$f(L_{\tilde{\mathcal{H}}}^*) \geq f(L_{\mathcal{H}}^*) - \rho \tag{2}$$

Since $L_{\tilde{\mathcal{L}}}^*$ is the optimal subset of size $M$ from the proxy agent set $\tilde{\mathcal{L}}$, and $L_{\tilde{\mathcal{H}}}^*$ is a feasible solution of size $M$ from $\tilde{\mathcal{L}}$, we have:

$$f(L_{\tilde{\mathcal{L}}}^*) \geq f(L_{\tilde{\mathcal{H}}}^*) \tag{3}$$

From the guarantees of the greedy algorithm for submodular function maximization (Nemhauser et al., 1978), if $L_{\tilde{\mathcal{L}}}^{greedy}$ is the subset of size $M$ returned by the greedy algorithm on $\tilde{\mathcal{L}}$, we have:

$$f(L_{\tilde{\mathcal{L}}}^{greedy}) \geq (1 - \frac{1}{e}) f(L_{\tilde{\mathcal{L}}}^*) \tag{4}$$

Combining inequalities equation 2, equation 3, and equation 4, we get:

$$f(L_{\tilde{\mathcal{L}}}^{greedy}) \geq (1 - \frac{1}{e}) f(L_{\tilde{\mathcal{L}}}^*) \geq (1 - \frac{1}{e}) f(L_{\tilde{\mathcal{H}}}^*) \geq (1 - \frac{1}{e})(f(L_{\mathcal{H}}^*) - \rho)$$

Using the definition of human coverage ratio $\gamma = \frac{f(L_{\mathcal{H}}^*)}{f(L_{\mathcal{L}}^*)}$, we have $f(L_{\mathcal{H}}^*) = \gamma \cdot f(L_{\mathcal{L}}^*)$. Substitution yields

$$f(L_{\tilde{\mathcal{L}}}^{greedy}) \geq (1 - \frac{1}{e})(\gamma \cdot f(L_{\mathcal{L}}^*) - \rho)$$

This gives us the performance guarantees for REPPOP$_{\text{mapped-1}}$ and REPPOP$_{\text{mapped-2}}$. $\qquad \square$

