# OpenReview forum: "Prompt Optimization Across Multiple Agents for Representing Diverse Human Populations"
_ICLR.cc/2026/Conference — Submitted to ICLR 2026_

### Official Review · Reviewer_SDgA · 2025-10-15

**Soundness:** 2
**Presentation:** 3
**Contribution:** 2
**Rating:** 6
**Confidence:** 3

**Summary:**

This paper addresses the challenge of Large Language Models (LLMs) producing homogeneous outputs that fail to represent the diversity of human populations. Instead of creating a single LLM agent, the authors propose a framework to construct a set of diverse agents that collectively capture the behavior of a target human population. Each agent's behavior is guided by a small set of human demonstrations through in-context learning. The core problem is selecting a representative set of agents from an exponentially large space, which the authors formulate as a submodular optimization problem. They develop several methods, such as $REPPOP_{demo}$ and $REPPOP_{mapped}$, that offer different trade-offs between computational complexity and performance. Experiments conducted in educational (EEDI dataset), opinion survey (OpinionQA dataset), and crowdsourcing (WikiArt dataset) domains show that the proposed methods construct agent sets that more effectively represent human populations compared to baselines. Furthermore, behavioral analysis demonstrates that these agents successfully reproduce the distinct behavior patterns and perspectives of the human subgroups they are designed to represent on new, unseen tasks.

**Strengths:**

1. A key strength is the innovative framing of the agent selection problem as a submodular optimization task. This provides a sound mathematical foundation for the problem, moving beyond simple heuristics and enabling the development of methods with theoretical performance guarantees.

2. The $REPPOP_{mapped}$ methods are a clever and crucial contribution. The heuristic to create a proxy agent for each human, thereby reducing an exponential search space to a linear one, is a well-justified move that makes the entire approach computationally feasible without resorting to purely random sampling.

3. The method is tested across diverse domains (education, opinion surveys, annotation) and multiple LLM families, demonstrating its model-agnostic effectiveness.

**Weaknesses:**

1. My primary concern lies with the WikiArt experiment, which is arguably the most complex and interesting testbed. The evaluation here relies entirely on "synthetic humans" generated by prompting another large language model (Gemma3-27B) with personality traits. This setup means the experiment is essentially testing how well the proposed framework can use a set of agents to model the behavior of another generative model, rather than the far more complex, noisy, and sometimes irrational behavior of actual humans. While this is a valid test of the optimization procedure itself, it significantly limits the claims that can be made about the method's effectiveness on genuine, open-ended human tasks. The success in this domain may not generalize to real-world scenarios where human responses are not as neatly conditioned on a few personality axes.

2. The core optimization objective, which minimizes the average distance from each human to their nearest agent, seems to overlook a crucial aspect: the population distribution. The metric $g(L)$ effectively rewards covering the entire behavioral space. However, it fails to account for how many people fall into certain behavioral clusters. For example, a population with an 80/20 split between two main opinion groups could be "optimally" represented by two agents, one for each group. This would imply a 50/50 distribution in the agent set, which is a gross misrepresentation of the actual population. This could be highly misleading for any downstream task that relies on statistical inference or simulating group dynamics.

3. Algorithm 1 describes a greedy, sequential process for building the M agents. It first finalizes the K demonstrations for the first agent, then moves on to the second, and so on. This approach introduces a strong path dependency. The choice of the very first agent significantly constrains the optimization landscape for all subsequent agents. If the first agent selected happens to cover a very broad but common part of the behavioral space, it might prevent the selection of a more nuanced set of specialist agents later on. A joint optimization over all M agents would be ideal, and while that is intractable, the current sequential approach may be prone to settling in a suboptimal local minimum for the entire set of agents.

4. The agents are stateless and reactive, lacking the memory and planning capabilities found in advanced agent frameworks like AutoGen [1] or social simulation platforms like OASIS [2]. This raises a critical question about their cognitive consistency: while they can mimic specific behaviors, they may fail to respond coherently when faced with novel situations that require a consistent worldview. They function less like a true simulated society and more like a collection of behavioral samplers.

[1] Wu, Qingyun, et al. "Autogen: Enabling next-gen LLM applications via multi-agent conversations." First Conference on Language Modeling. 2024.

[2] Yang, Ziyi, et al. "Oasis: Open agent social interaction simulations with one million agents." arXiv preprint arXiv:2411.11581 (2024).

**Questions:**

Have the authors considered the trade-offs between their ensemble approach and an alternative using a single, powerful SOTA model? For instance, could one use a model like GPT-4 with a rich, dynamically-retrieved context (via RAG) to simulate diverse individuals with potentially greater cognitive consistency? I'm curious what the authors see as the key advantages of their specialist ensemble method over this kind of 'single generalist' approach.

---

> ### Author Response · Authors · 2025-11-21
>
> Thank you for carefully reviewing our paper! We greatly appreciate your feedback. Below, we address the reviewer's concerns regarding generalization to open-ended human tasks, discuss extensions for statistical inference and group dynamics simulation and clarify the rationale and effectiveness of our sequential greedy approach for joint optimization. We also discuss potential extensions to advanced agent frameworks with memory and planning, and provide a detailed comparison of our specialist ensemble approach versus single generalist models with RAG. Please find our detailed responses to the reviewer's comments below.
>
> -----
> **Generalization to open-ended human tasks**
>
> The main reason for using LLMs to generate annotation data in the WikiArt experiment was the absence of annotator-level data in the relevant literature for this domain (as mentioned in Section 5.1). This data is crucial for our method, as it requires observing multiple demonstrations from the same individual across different tasks to construct meaningful behavioral embeddings. Due to resource and time constraints, we were unable to conduct a new human study and therefore used LLMs as annotators in this domain.
>
> We believe that our experimental results on the EEDI and OpinionQA datasets, which use real human data, provide stronger evidence of our framework's effectiveness in capturing genuine human behavior. The WikiArt dataset serves as a complementary and preliminary experiment in annotating visual art. We will include this limitation and propose future work on validating our framework with other human annotation datasets in the updated paper.
>
> -----
> **Statistical inference or simulating group dynamics.**
>
> Our objective function prioritizes behavioral coverage over the proportional representation of subpopulations. This is indeed a deliberate design choice, and we appreciate the opportunity to clarify the rationale and discuss its implications.
>
> Our goal is to construct a set of agents that can collectively handle the diverse range of behaviors observed in a human population. In this framing, the key question is: "For each human, can we find an agent that behaves similarly to them?". An example use case of this coverage-based objective is educational applications, where teachers need agents representing students with different misconceptions or skill levels to practice instructional strategies, regardless of how common each misconception is.
>
> We agree with the reviewer that for tasks requiring statistical inference or simulating group dynamics where the goal is to estimate population-level statistics or model collective behavior, proportional representation is important. A straightforward extension that addresses this concern is to sample humans according to their population distribution and use their corresponding nearest agent. Specifically, when simulating group dynamics or performing statistical inference: (1) sample a human $h$ from the population according to the actual distribution, (2) find the nearest agent $\ell^* = \arg\min_{\ell \in L} \text{dist}(e_h, e_\ell)$, and (3) use agent $\ell^*$ as a proxy for human $h$. This approach preserves population proportions during agent usage while maintaining the representation optimization objective.
>
> Alternatively, one could modify the objective function to incorporate population weights which reflect the importance or frequency of human $h$'s behavioral profile. We believe both directions are valuable for extending our framework to applications requiring population-level inference.
>
> We will add a discussion of this limitation and potential extensions to the updated paper.

---

> ### Author Response · Authors · 2025-11-21
>
> -----
> **Joint optimization over all M agents**
>
> We thank the reviewer for this suggestion. We clarify that the sequential greedy approach is not a limitation specific to our method, but a fundamental characteristic of all practical algorithms for submodular maximization problems. As we prove in Theorem 1, the underlying optimization problem is NP-hard, making exact joint optimization computationally intractable. The standard approach in the submodular optimization literature is to use greedy algorithms, which provide theoretical guarantees ($(1 - 1/e)$-approximation guarantee) despite their sequential nature.
>
> The reviewer has a concern about the first agent potentially covering "a very broad but common part of the behavioral space" and thus preventing the selection of specialist agents later. However, our objective function directly mitigates this issue. The marginal gain of adding agent $l$ at iteration $i$ (which corresponds to the marginal gain of the submodular function that we maximize) is: $\Delta(l | L_{i-1}) = g(L_{i-1}) - g(L_{i-1} \cup \{l\})$. This measures how much agent $l$ reduces the representation gap given the agents already selected. Crucially, if the first agent covers a broad region, humans in that region will already be well-represented (i.e., they are already close to an existing agent). For such humans, adding another agent nearby provides little additional benefit, resulting in a small marginal gain. The greedy algorithm selects the agent with maximum marginal gain at each iteration, which will naturally be one that covers currently underrepresented behavioral regions where the marginal gain is larger. In Figure 4(b), we demonstrate that agents capture subpopulations with specific demographics, confirming that our method successfully selects diverse specialists rather than redundant generalists.
>
>
> -----
> **Advanced agent frameworks with memory and planning**
>
> We agree that incorporating memory and planning capabilities, as in frameworks like AutoGen or OASIS, would be a natural and valuable extension of our work, particularly for applications requiring long-term social dynamics or multi-step reasoning with persistent beliefs. Thoroughly testing such applications, however, is a complex and resource-intensive endeavor that goes beyond the scope of the current work. We will include a discussion of this potential extension to the updated paper.
>
> -----
> **Single generalist' approach with a SOTA model**
>
> We thank the reviewer for this question about trade-offs between our specialist ensemble approach and a single generalist model with RAG. We address both the cognitive consistency concern and the key advantages of our method.
>
> We argue that our persistent specialist agents provide greater behavioral consistency than a single generalist with dynamic RAG. A single model with RAG retrieves different demonstrations per query based on semantic similarity, which can lead to inconsistent behavioral patterns across related tasks. In contrast, our agents maintain fixed demonstration sets, ensuring that each agent responds consistently to similar scenarios. Our experiments (Section 5.4) demonstrate that agents maintain stable behavioral patterns on held-out test tasks, with agents representing specific subpopulations (e.g., students with particular skill profiles, or individuals with distinct political views) consistently across different questions.
>
> The key advantage of our specialist ensemble over a single generalist is explicit population coverage. A single powerful model, even with RAG, must balance representing diverse subpopulations within a single set of parameters, which can lead to averaging effects that obscure minority viewpoints. Our ensemble explicitly constructs $M$ distinct agents, each optimized to represent a specific subpopulation, ensuring comprehensive coverage of the behavioral space. This is particularly important for population simulation where we need to understand how different groups respond.
>
> While RAG performs task-conditioned retrieval, our method optimizes agent selection globally over the entire population. This ensures that the agent set collectively covers the entire population's behavioral space, rather than optimizing for individual tasks. Global optimization is essential for population-level simulation where agents must represent consistent behavioral identities across diverse scenarios. We acknowledge that a single generalist with RAG offers flexibility through dynamic adaptation, which may be advantageous for applications requiring task-specific optimization. However, for population simulation applications where consistent behavioral representation and comprehensive population coverage are priorities, our specialist ensemble approach provides advantages.
>
> We will include this comparison in the updated paper to better highlight the key advantages of our approach over RAG for population simulation applications.

---

> > ### Comment · Reviewer_SDgA · 2025-11-27
> >
> > Thank you for the rebuttal. I have read the response and decided to maintain my original score.

---

> > > ### Author Response · Authors · 2025-11-28
> > >
> > > We sincerely thank the reviewer for their insightful feedback and engagement during the discussion period. We will incorporate the reviewer's feedback and our responses in the updated paper.

---

### Official Review · Reviewer_6gHd · 2025-11-01

**Soundness:** 2
**Presentation:** 2
**Contribution:** 1
**Rating:** 2
**Confidence:** 3

**Summary:**

The paper introduces a submodular optimization framework to select a set of in-context-conditioned LLM agents that collectively represent human population diversity. Experiments in crowdsourcing and education show improved alignment with human behaviors. Limitations include reliance on prompting, lack of prompt-order analysis, and limited real-world evaluation.

**Strengths:**

The topic is important: LLMs are increasingly used as proxies for human behavior, yet current models tend to collapse toward homogeneous responses.

**Weaknesses:**

- Theorem 1 (NP-hardness of subset selection) and Proposition 1 (Submodularity of f(L)). The NP-hardness of this subset selection objective and submodularity property of f(L) is well established in prior work (e.g., k-center, facility-location), so the full proof seems unnecessary; a short reference would make the paper more focused on main novelty.
- The proposed framework closely mirrors prior work [1], particularly in using submodular optimization for in-context example selection. The contribution appears incremental rather than conceptually new.
- The paper does not define or explain the Representation Error metric, making it difficult to interpret results or compare performance against existing baselines.
- The paper lacks performance or conceptual comparisons with strong existing baselines [2–4]. Specifically, Bui et al. [2] propose a contextual mixture-of-personas model that aligns LLM behavior with human populations using persona descriptions and in-context examples. Choi & Li [3] introduce PiCLe, an in-context learning framework designed to elicit diverse human-like behaviors, while Yu et al. [4] present a prompting-based approach to generate diverse and bias-controlled data.
- Experimenting on OpinionQA is questionable, see [5]. Dominguez-Olmedo et al. (2024) show that survey-style prompting of LLMs yields unreliable signals: responses are dominated by ordering and labeling biases, and once controlled, model outputs become nearly uniform and high-entropy regardless of size or tuning. As a result, survey-based “alignment” measures mainly capture random or positional effects rather than genuine human-like diversity. Using OpinionQA therefore, risks over-interpreting noise as meaningful representativeness, weakening the validity of the claimed behavioral diversity results.

References:
- [1] Ji, Baijun, et al. "Submodular-based in-context example selection for LLMs-based machine translation." Proceedings of the 2024 Joint International Conference on Computational Linguistics, Language Resources and Evaluation (LREC-COLING 2024). 2024.
- [2] Bui, Ngoc, et al. "Mixture-of-personas language models for population simulation." arXiv preprint arXiv:2504.05019 (2025).
- [3] Choi, Hyeong Kyu, and Yixuan Li. "Picle: Eliciting diverse behaviors from large language models with persona in-context learning." arXiv preprint arXiv:2405.02501 (2024).
- [4] Yu, Yue, et al. "Large language model as attributed training data generator: A tale of diversity and bias." Advances in neural information processing systems 36 (2023): 55734-55784.
- [5] Dominguez-Olmedo, Ricardo, Moritz Hardt, and Celestine Mendler-Dünner. "Questioning the survey responses of large language models." Advances in Neural Information Processing Systems 37 (2024): 45850-45878.

**Questions:**

Please see weaknesses

---

> ### Author Response · Authors · 2025-11-21
>
> Thank you for carefully reviewing our paper! We greatly appreciate your feedback. Below, we provide a detailed comparison to the work of Baijun et al. highlighting fundamental differences in problem formulation and objectives. We also clarify the definition of the Representation Error metric, add comprehensive conceptual comparison and additional experimental evaluation compared to prior work (AttrPrompt, PICLe, and Mixture-of-Personas). In addition, we address concerns about survey-based alignment on OpinionQA with evidence of meaningful behavioral diversity, and refine our presentation of theoretical results. Please find our detailed responses to the reviewer's comments below.
>
> -----
> **Differences compared to the work of Baijun et al.**
>
> We thank the reviewer for pointing out this related work. While both works utilize submodular optimization, the problem formulation, objective function, computational structure, and application domain are fundamentally different. Below we provide a detailed comparison:
>
> (1) Problem Formulation and Objective:
> - Baijun et al.: Select examples to maximize translation *quality* (BLEU score) for a *single* translation model. The objective is task-specific performance optimization.
> - Our work: Select a *set of $M$ agents* to minimize the *representation gap* across a human population. We do not optimize for task correctness, but rather for behavioral coverage and diversity.
>
> (2) Output Structure:
> - Baijun et al.: Constructs a *single* prompt (or per-instance prompts) for *one* model. The selection space is over individual examples.
> - Our work: Constructs an *ensemble of $M$ distinct agents*, where each agent is defined by a set of $K$ demonstrations. The selection space is over agents, and we solve a *set selection problem over agents*, not examples. This is a multi-agent coverage problem, not single-agent optimization.
>
> (3) Computational Structure:
> - Baijun et al.: Standard submodular maximization over a fixed example pool. The complexity is $\mathcal{O}(M \cdot |\mathcal{E}|)$ for selecting $M$ examples.
> - Our work: We face an combinatorially large agent space $|\mathcal{L}| = \binom{|\mathcal{D}|}{K}$. One of our key contributions is the *human-mapped proxy construction* (Section 4.2), which reduces the search space from combinatorial to linear ($|\mathcal{H}|$) while preserving theoretical guarantees.
>
> (4) Evaluation:
> - Baijun et al.: Machine translation with evaluation via BLEU score (a correctness metric).
> - Our work: Population simulation with evaluation via representation gap. We explicitly aim to capture the *full spectrum* of human behaviors, including incorrect answers, minority opinions, and diverse perspectives.
>
>
> In summary, while both works use submodular optimization, Baijun et al. optimize examples for a *single* model's performance, whereas we solve a fundamentally different problem of constructing an *ensemble* of agents that collectively represent a human population. We believe our contributions are specific to the multi-agent coverage problem and represent a novel application of submodular optimization. We will cite and discuss this work in our revised related work section to clarify these distinctions.
>
> -----
> **Definition of Representation Error metric**
>
> We thank the reviewer for raising this question. The representation error is the term we use in Section 5 to refer to the normalized representation gap. Specifically, it is the representation gap normalized by the embedding dimensionality to make results comparable across datasets with different embedding dimensions. We normalize by $d$ for EEDI (binary vectors with L1 distance) and by $\sqrt{d}$ for OpinionQA and WikiArt (continuous vectors with L2 distance), where $d$ is the embedding dimension. The underlying representation gap $g(L)$ is the metric defined in Section 3. It measures the average distance between each human $h$ and their closest agent $l \in L$. We will explicitly state this in the updated paper.

---

> > ### Author Response · Authors · 2025-11-21
> >
> > -----
> > **Conceptual comparison to prior work**
> >
> > Below we provide a detailed comparison of our methods to prior work. In particular, we compare our methods to AttrPrompt (Yu et al.'23), PICLe (Choi et al. '24), and Mixture-of-Personas (Bui et al.'25). In summary, our method has several key advantages over these baselines. First, our method works directly with raw behavioral data (demonstrations), thus reducing potential bias/stereotyping. Second, our method does not require access to LLM logits or fine-tuning. Finally, our submodular optimization framework provides theoretical guarantees on representation gap.
> >
> > *Table: Conceptual comparison to prior work*
> > |  | **AttrPrompt (Yu et al.'23)** | **PICLe (Choi et al. '24)** | **Mixture-of-Personas (Bui et al.'25)** | **Ours** |
> > |:-----------|:---------------------------|:-------------------------------------|:----------------------|:---------------------------|
> > | **Main Goal** | Generate diverse training data for text classification | Elicit specific persona from LLM | Align LLM responses with target population distribution | Representative agent set selection |
> > | **Approach** | Attributed prompts with random attribute combinations | In-context learning with likelihood-ratio example selection | Two-level hierarchical mixture model with personas + exemplars | Submodular optimization over agent space |
> > | **Optimization Objective** | Implicit: maximize attribute diversity in generated data | Maximize likelihood ratio for target persona | Maximum likelihood estimation on population observations | Minimize representation gap: avg. distance from each human to nearest agent |
> > | **Number of Agents/Outputs** | N/A | Single agent eliciting one persona (1-to-1 mapping) | Single agent with probabilistic mixture of $M$ personas | $M$ persistent agents, each representing a subpopulation |
> > | **Uses persona/attribute** | Attribute | Persona | Persona | No |
> > | **Requires Training/Fine-tuning** | No | Yes | Yes | No |
> > | **Requires LLM Output Logits** | No | Yes | Yes | No |
> > | **Population Representation** | Implicit via attribute diversity | Single persona (not population-level) | Probabilistic mixture ensures population alignment | Explicit via representation gap minimization |
> > | **Theoretical Guarantees** | No | No | No | Yes |
> >
> > We will update the paper to include this additional conceptual comparison to highlight the key differences between our work and prior work.

---

> > > ### Author Response · Authors · 2025-11-21
> > >
> > > -----
> > > **Experimental comparison to baselines from prior work (AttrPrompt, PICLe, Mixture-of-Personas)**
> > >
> > > We implemented the 3 baselines mentioned above and adapted them to our setting to provide additional experimental comparison. We conducted these experiments on the two real-world datasets (EEDI and OpinionQA) using Gemma-12B to align with our main experiment in the paper. The results are reported in the table below.
> > >
> > > *Table: Representation Error (mean $\pm$ standard error) across different numbers of demonstrations $K$. We report for $M=10$ agents. The best performance (lowest error) is highlighted in bold.*
> > > | Method | EEDI $(K=1)$ | EEDI $(K=3)$ | EEDI $(K=5)$ | OpinionQA $(K=1)$ | OpinionQA $(K=3)$ | OpinionQA $(K=5)$ |
> > > | :--- | :---: | :---: | :---: | :---: | :---: | :---: |
> > > | $\rm{Single}$ | 0.41 $\pm$ 0.03 | 0.40 $\pm$ 0.03 | 0.36 $\pm$ 0.04 | 1.09 $\pm$ 0.00 | 1.10 $\pm$ 0.01 | 1.09 $\pm$ 0.02|
> > > | $\rm{Random}$ | 0.31 $\pm$ 0.02 | 0.34 $\pm$ 0.01 | 0.37 $\pm$ 0.02 | 0.99 $\pm$ 0.02 | 0.97 $\pm$ 0.01 | 0.99 $\pm$ 0.01 |
> > > | $\rm{K-Medoids}$ | 0.31 $\pm$ 0.01 | 0.35 $\pm$ 0.03 | 0.36 $\pm$ 0.01 | 0.98 $\pm$ 0.01 | 1.00 $\pm$ 0.01 | 0.98 $\pm$ 0.01 |
> > > |    |
> > > | $\rm{AttrPrompt}$ (Yu et al.) | 0.36 $\pm$ 0.01 | 0.38 $\pm$ 0.01 | 0.41 $\pm$ 0.01 | 0.97 $\pm$ 0.01 | 0.97 $\pm$ 0.01 | 0.96 $\pm$ 0.01 |
> > > | $\rm{PICLe}$ (Choi et al.) | 0.31 $\pm$ 0.00 | 0.34 $\pm$ 0.01 | 0.33 $\pm$ 0.01 | 1.03 $\pm$ 0.01 | 0.95 $\pm$ 0.01 | 0.99 $\pm$ 0.00 |
> > > | $\rm{Mixture-of-Personas}$ (Bui et al.) | 0.32 $\pm$ 0.02 | 0.34 $\pm$ 0.01 | 0.33 $\pm$ 0.01 | **0.96 $\pm$ 0.01** | 0.96 $\pm$ 0.01 | 0.97 $\pm$ 0.01 |
> > > |    |
> > > | $\rm{SampleGreedy}$ | 0.31 $\pm$ 0.02 | 0.32 $\pm$ 0.01 | **0.31 $\pm$ 0.01** | 0.99 $\pm$ 0.01 | 0.93 $\pm$ 0.01 | 0.92 $\pm$ 0.00 |
> > > | $\rm{RepPop}_{\rm{demo}}$ (ours) | 0.30 $\pm$ 0.01 | **0.31 $\pm$ 0.01** | **0.31 $\pm$ 0.01** | 1.00 $\pm$ 0.01 | 0.95 $\pm$ 0.01 | 0.93 $\pm$ 0.01 |
> > > | $\rm{RepPop}_{\rm{mapped}-1}$ (ours) | 0.31 $\pm$ 0.01 | 0.33 $\pm$ 0.01 | 0.34 $\pm$ 0.01 | 0.99 $\pm$ 0.00 | 0.92 $\pm$ 0.01 | 0.92 $\pm$ 0.02 |
> > > | $\rm{RepPop}_{\rm{mapped}-2}$ (ours) | **0.29 $\pm$ 0.02** | **0.31 $\pm$ 0.01** | **0.31 $\pm$ 0.01** | 0.99 $\pm$ 0.00 | **0.89 $\pm$ 0.02** | **0.91 $\pm$ 0.01** |
> > >
> > > In summary, while these additional baselines outperform our simpler baseline methods in some settings, they still underperform our proposed methods overall. The only exception is Mixture-of-Personas on OpinionQA with $K=1$, where it slightly outperforms our methods. In all other cases, our methods, especially $\rm{RepPop}_{\rm{mapped}-2}$, achieve the lowest representation error.
> > >
> > > We will update the paper to include this additional baseline experiment to further strengthen our results.

---

> > > > ### Author Response · Authors · 2025-11-21
> > > >
> > > > -----
> > > > **Survey-based "alignment" on OpinionQA**
> > > >
> > > > We thank the reviewer for raising this important point. The core critique given by Dominguez-Olmedo et al. was that extracting next-token probabilities from LLMs for survey questions yields unreliable signals dominated by ordering and labeling biases. This often results in uniform, high-entropy responses after control.
> > > >
> > > > We first clarify that, instead of extracting next-token probabilities over answer choice tokens (A, B, C, D), our method generates full text responses and then parses the generated text to extract the selected answer choice. While ordering and labeling biases may still affect text generation, the full-text generation process allows the model to consider the semantic content of answer choices rather than relying solely on positional token probabilities. Future work on evaluating the impact of ordering and labeling biases on our methods would be valuable.
> > > >
> > > > Next, we provide evidence that our agents capture meaningful behavioral diversity rather than noise. First, if our agents were merely capturing random entropy or ordering bias, the optimized set would fail to generalize to held-out questions. The fact that the representation error decreases consistently on the test set (as shown in Figure 3 and Figure 10-15) demonstrates that we are capturing stable underlying signals that transfer across question sets. Second, in Section 5.4 and specifically Figure 4(b), we demonstrate that agents constructed on training questions exhibit distinct and interpretable political opinion distributions on held-out test questions. Notably, the political opinion distributions produced by agents on test questions match the aggregated demographics of the humans they represent, despite demographic metadata never being used in agent construction. This alignment validates that our approach generates meaningful behavioral diversity, which would not occur if responses were dominated by random effects or consistent positional biases.
> > > >
> > > > Finally, OpinionQA is only one of three diverse datasets validating our framework. The consistent reduction in representation gap across all three domains, including open-ended annotations (WikiArt), demonstrates that our methods capture meaningful behavioral diversity.
> > > >
> > > > -----
> > > > **A short reference to proofs of Theorem 1 and Proposition 1 would make the paper more focused on main novelty.**
> > > >
> > > > We thank the reviewer for this suggestion. In our submission, we included only brief proof sketches in the main paper (Section 4.1) citing the connection to facility location problems, with full formal proofs provided in the appendix. While the underlying NP-hardness and submodularity results build on classical facility location theory, the connection to our specific problem is novel due to several factors. First, the agent space $\mathcal{L}$ is defined by combinatorial choices of K demonstrations. Second, agent embeddings $\mathbf{e}_l$ are obtained through LLM inference rather than being predetermined. Third, we minimize distances in a learned behavioral space. Nevertheless, we will consider presenting Theorem 1 and Proposition 1 as two propositions in our updated paper, and focus more on Theorem 2 regarding our methods.

---

> ### Comment · Reviewer_6gHd · 2025-11-28
> **Response to rebuttal**
>
> Thanks for author's rebuttal.
> All my concerns have been addressed. I will raise my overall rating to 4.

---

> > ### Author Response · Authors · 2025-11-28
> >
> > We sincerely thank the reviewer for their helpful feedback and engagement during the discussion period. We are glad that our responses have fully addressed the reviewer's concerns and appreciate the score increase. We will incorporate our discussion and the additional experiments into the updated version of the paper.

---

### Official Review · Reviewer_HSz7 · 2025-11-03

**Soundness:** 1
**Presentation:** 3
**Contribution:** 2
**Rating:** 4
**Confidence:** 3

**Summary:**

The paper frames population modeling as selecting a small set of in-context LLM `agents' to cover human behaviors by minimizing a representation gap in an embedding space. They introduce a submodular objective and propose a practical greedy solution to find a subset of agents that are representative of the human population. They report gains over some standard selection baselines across multiple tasks.

**Strengths:**

- The paper studies an important and emerging problem that has received significant attention recently.
- The authors use broad empirical evaluation across several important datasets.
- Clear ablations/visualizations that help interpret coverage and trade-offs.

**Weaknesses:**

- Motivation for the formulation is unclear. Why should minimizing training-task embedding distances ensure alignment on unseen tasks?
- Unclear construction of agent embeddings e_l an where exactly LLM's outputs enter the objective. Do you use LLM's logits to compute e_l, or does it solely depend on the in-context demonstrations?
- Missing comparisons with mixture-of-agents/personas methods (e.g., PICLe (Choi et al. PICLe: Eliciting Diverse Behaviors from Large Language Models with Persona In-Context Learning); mixture-of-personas LMs (Bui et al. Mixture-of-Personas Language Models for Population Simulation)

**Questions:**

- How exactly is e_l computed for each dataset? Does it depend on LLM's output or just the in-context example only?
- Where does LLM’s output come into place in the formulation (1)? Is it e_l? Each LLM has a different bias. How to mitigate these biases for a specific LLM?
- Scalability with M: what if we want to analyze a larger population where M is large? Does your method scale well with M and K?

---

> ### Author Response · Authors · 2025-11-21
>
> Thank you for carefully reviewing our paper! We greatly appreciate your feedback. Below, we clarify the motivation for our formulation, provide explanations and concrete examples of agent embedding construction. We also add comprehensive conceptual comparison and additional experimental evaluation compared to prior work (AttrPrompt, PICLe, and Mixture-of-Personas). In addition, we address concerns about LLM bias mitigation, and answer questions about scalability analysis. Please find our detailed responses to the reviewer's comments below.
>
> -----
> **Motivation for the formulation and transfer to unseen tasks.**
>
> Our formulation minimizes the representation gap on training tasks because the embeddings capture behaviors of humans/agents, which we assume are consistent and transfer to unseen tasks. We expect this transfer assumption to hold due to two reasons: (1) training and test data come from the same population of humans, (2) tasks are sampled from the same distribution. Under these conditions, minimizing the representation gap on training tasks ensures that the selected agents faithfully represent the population's behavioral diversity, which carries over to held-out test sets. In Appendix C, we empirically demonstrate this on EEDI, OpinionQA, and WikiArt, where the reduction in representation gap on training sets transfers to their respective held-out test sets.
>
> -----
> **The construction of agent embeddings $e_l$**
>
> In formulation (1), the LLM agent's outputs enter the objective through the agent embedding $e_l$. We keep the embedding notation general in our setup to accommodate domain-specific types of embeddings in each dataset/domain. Specifically, the agent embedding $e_l$ in formulation (1) is derived from the LLM's generated answers to a set of tasks. The LLM generates responses based on its in-context demonstrations; these responses are then embedded into the same behavioral representation space as human embeddings. In essence, agent embeddings are computed in the same way as human embeddings. The in-context demonstrations only affect the embedding indirectly by steering the agent's answers. We do not use token-level logits anywhere in constructing embeddings or in the objective.
>
> We discussed the specific types of embeddings used in each dataset in Section 5.1. Below we provide some examples to clarify how we constructed agent embeddings.
> - EEDI dataset: Each student/agent is represented by a binary embedding vector indicating their correct and incorrect answers to the math questions. For example, [1,0,...,1] represents [correct, incorrect,…,correct]. This vector summarizes the behavior of the student/agent on a set of math questions.
> - OpinionQA dataset: For each question, answer choices are mapped to ordinal values and normalized to the range [−1,1]. Each human is represented by a vector embedding of their responses. For example, [-1, 0.5,...,1] represents ['worse', 'Increased somewhat', …, 'Greatly expand on current government services']. This vector summarizes the behavior of the person/agent on a set of survey questions.
> - WikiArt dataset: For each annotator, we concatenate their answers across tasks into a single prompt (cf. Figure 8 in the paper), pass it through a language model (Gemma3-12B), and extract the last hidden state with mean pooling to obtain the embedding. We then reduce the dimensionality to 64 using PCA. The resulting continuous embedding vector (e.g., [0.1032, -0.2211,..., -0.7715]) summarizes the behavior of the human annotator/agent on a set of tasks.
>
> We will update the paper to clarify the construction of agent embeddings and provide these concrete examples.

---

> > ### Author Response · Authors · 2025-11-21
> >
> > -----
> > **Conceptual comparison to prior work**
> >
> > Below we provide a detailed comparison of our methods to prior work. In particular, we compare our methods to AttrPrompt (Yu et al.'23), PICLe (Choi et al. '24), and Mixture-of-Personas (Bui et al.'25). In summary, our method has several key advantages over these baselines. First, our method works directly with raw behavioral data (demonstrations), thus reducing potential bias/stereotyping. Second, our method does not require access to LLM logits or fine-tuning. Finally, our submodular optimization framework provides theoretical guarantees on representation gap.
> >
> > *Table: Conceptual comparison to prior work*
> > |  | **AttrPrompt (Yu et al.'23)** | **PICLe (Choi et al. '24)** | **Mixture-of-Personas (Bui et al.'25)** | **Ours** |
> > |:-----------|:---------------------------|:-------------------------------------|:----------------------|:---------------------------|
> > | **Main Goal** | Generate diverse training data for text classification | Elicit specific persona from LLM | Align LLM responses with target population distribution | Representative agent set selection |
> > | **Approach** | Attributed prompts with random attribute combinations | In-context learning with likelihood-ratio example selection | Two-level hierarchical mixture model with personas + exemplars | Submodular optimization over agent space |
> > | **Optimization Objective** | Implicit: maximize attribute diversity in generated data | Maximize likelihood ratio for target persona | Maximum likelihood estimation on population observations | Minimize representation gap: avg. distance from each human to nearest agent |
> > | **Number of Agents/Outputs** | N/A | Single agent eliciting one persona (1-to-1 mapping) | Single agent with probabilistic mixture of $M$ personas | $M$ persistent agents, each representing a subpopulation |
> > | **Uses persona/attribute** | Attribute | Persona | Persona | No |
> > | **Requires Training/Fine-tuning** | No | Yes | Yes | No |
> > | **Requires LLM Output Logits** | No | Yes | Yes | No |
> > | **Population Representation** | Implicit via attribute diversity | Single persona (not population-level) | Probabilistic mixture ensures population alignment | Explicit via representation gap minimization |
> > | **Theoretical Guarantees** | No | No | No | Yes |
> >
> > We will update the paper to include this additional conceptual comparison to highlight the key differences between our work and prior work.

---

> > > ### Author Response · Authors · 2025-11-21
> > >
> > > -----
> > > **Experimental comparison to baselines from prior work (AttrPrompt, PICLe, Mixture-of-Personas)**
> > >
> > > We implemented the 3 baselines mentioned above and adapted them to our setting to provide additional experimental comparison. We conducted these experiments on the two real-world datasets (EEDI and OpinionQA) using Gemma-12B to align with our main experiment in the paper. The results are reported in the table below.
> > >
> > > *Table: Representation Error (mean $\pm$ standard error) across different numbers of demonstrations $K$. We report for $M=10$ agents. The best performance (lowest error) is highlighted in bold.*
> > > | Method | EEDI $(K=1)$ | EEDI $(K=3)$ | EEDI $(K=5)$ | OpinionQA $(K=1)$ | OpinionQA $(K=3)$ | OpinionQA $(K=5)$ |
> > > | :--- | :---: | :---: | :---: | :---: | :---: | :---: |
> > > | $\rm{Single}$ | 0.41 $\pm$ 0.03 | 0.40 $\pm$ 0.03 | 0.36 $\pm$ 0.04 | 1.09 $\pm$ 0.00 | 1.10 $\pm$ 0.01 | 1.09 $\pm$ 0.02|
> > > | $\rm{Random}$ | 0.31 $\pm$ 0.02 | 0.34 $\pm$ 0.01 | 0.37 $\pm$ 0.02 | 0.99 $\pm$ 0.02 | 0.97 $\pm$ 0.01 | 0.99 $\pm$ 0.01 |
> > > | $\rm{K-Medoids}$ | 0.31 $\pm$ 0.01 | 0.35 $\pm$ 0.03 | 0.36 $\pm$ 0.01 | 0.98 $\pm$ 0.01 | 1.00 $\pm$ 0.01 | 0.98 $\pm$ 0.01 |
> > > |    |
> > > | $\rm{AttrPrompt}$ (Yu et al.) | 0.36 $\pm$ 0.01 | 0.38 $\pm$ 0.01 | 0.41 $\pm$ 0.01 | 0.97 $\pm$ 0.01 | 0.97 $\pm$ 0.01 | 0.96 $\pm$ 0.01 |
> > > | $\rm{PICLe}$ (Choi et al.) | 0.31 $\pm$ 0.00 | 0.34 $\pm$ 0.01 | 0.33 $\pm$ 0.01 | 1.03 $\pm$ 0.01 | 0.95 $\pm$ 0.01 | 0.99 $\pm$ 0.00 |
> > > | $\rm{Mixture-of-Personas}$ (Bui et al.) | 0.32 $\pm$ 0.02 | 0.34 $\pm$ 0.01 | 0.33 $\pm$ 0.01 | **0.96 $\pm$ 0.01** | 0.96 $\pm$ 0.01 | 0.97 $\pm$ 0.01 |
> > > |    |
> > > | $\rm{SampleGreedy}$ | 0.31 $\pm$ 0.02 | 0.32 $\pm$ 0.01 | **0.31 $\pm$ 0.01** | 0.99 $\pm$ 0.01 | 0.93 $\pm$ 0.01 | 0.92 $\pm$ 0.00 |
> > > | $\rm{RepPop}_{\rm{demo}}$ (ours) | 0.30 $\pm$ 0.01 | **0.31 $\pm$ 0.01** | **0.31 $\pm$ 0.01** | 1.00 $\pm$ 0.01 | 0.95 $\pm$ 0.01 | 0.93 $\pm$ 0.01 |
> > > | $\rm{RepPop}_{\rm{mapped}-1}$ (ours) | 0.31 $\pm$ 0.01 | 0.33 $\pm$ 0.01 | 0.34 $\pm$ 0.01 | 0.99 $\pm$ 0.00 | 0.92 $\pm$ 0.01 | 0.92 $\pm$ 0.02 |
> > > | $\rm{RepPop}_{\rm{mapped}-2}$ (ours) | **0.29 $\pm$ 0.02** | **0.31 $\pm$ 0.01** | **0.31 $\pm$ 0.01** | 0.99 $\pm$ 0.00 | **0.89 $\pm$ 0.02** | **0.91 $\pm$ 0.01** |
> > >
> > > In summary, while these additional baselines outperform our simpler baseline methods in some settings, they still underperform our proposed methods overall. The only exception is Mixture-of-Personas on OpinionQA with $K=1$, where it slightly outperforms our methods. In all other cases, our methods, especially $\rm{RepPop}_{\rm{mapped}-2}$, achieve the lowest representation error.
> > >
> > > We will update the paper to include this additional baseline experiment to further strengthen our results.
> > >
> > > -----
> > > **Each LLM has a different bias. How to mitigate these biases for a specific LLM?**
> > >
> > > We agree that each LLM has its own biases that can make them unrepresentative of certain population groups. Our method does not directly mitigate biases within a single LLM, but rather constructs a set of LLM agents  that collectively are more representative and less biased than the single LLM agent. This is achieved through two key mechanisms. First, each agent is created by conditioning the LLM on demonstrations from actual humans in the target population, which steers individual agents toward behaviors that reflect different human subpopulations. Second, our methods explicitly selects a set of agents to minimize the representation gap between the human population and the set of representative agents. The optimization process ensures that the selected set of agents collectively covers the heterogeneous behavioral space of the population. We note that our framework is model-agnostic. In our experiments (Table 3), we show that our methods are robust to the choice of underlying LLM in terms of both model family and model size, demonstrating that the methods constructing a representative set of agents works regardless of the specific LLM used.

---

> > > > ### Author Response · Authors · 2025-11-21
> > > >
> > > > -----
> > > > **Scalability with M and K**
> > > >
> > > > Our complexity analysis, summarized in Table 1, shows that our proposed methods *scale linearly* with both the number of agents $M$ and the context size $K$. In contrast, the standard $\rm{Greedy}$ and $\rm{SampleGreedy}$ methods remain computationally intractable due to their exponential dependence on $K$, scaling as $\mathcal{O}(M \cdot |\mathcal{H}| \cdot (|\mathcal{T}| \cdot |\mathcal{H}|)^K)$ and $\mathcal{O}(M \cdot |\mathcal{H}| \cdot \psi \cdot (|\mathcal{T}| \cdot |\mathcal{H}|)^K)$ respectively. We eliminate this exponential dependence. Our method $RepPop_{demo}$ achieves linear scalability with time complexity $\mathcal{O}(M \cdot K \cdot |\mathcal{T}| \cdot |\mathcal{H}|^2)$. Our human-mapped methods, $RepPop_{mapped-1}$ and $RepPop_{mapped-2}$, further improve efficiency by isolating agent construction from selection. Specifically, $RepPop_{mapped-1}$ has complexity $\mathcal{O}(K \cdot |\mathcal{H}| + M \cdot |\mathcal{H}|^2)$ (random sampling + greedy selection), while $RepPop_{mapped-2}$ has complexity $\mathcal{O}(K \cdot |\mathcal{T}| \cdot |\mathcal{H}| + M \cdot |\mathcal{H}|^2)$ (greedy demonstration selection + greedy selection). By reducing the candidate pool from the combinatorially large agent space $|\mathcal{L}| = \binom{|\mathcal{D}|}{K}$ to size $|\mathcal{H}|$, our methods remain computationally tractable even as $M$ and $K$ increase.

---

> > > > > ### Author Response · Authors · 2025-11-28
> > > > >
> > > > > We sincerely thank the reviewer for their constructive feedback. We have provided answers to the reviewer's comments and questions. We have also added conceptual comparisons as well as additional experimental results against three existing methods, including those suggested by the reviewer. We hope our responses are helpful in improving your rating. As there is still time for discussion, please let us know if there is anything else we can do to fully address your concerns and further strengthen our work.

---

### Author Response · Authors · 2025-12-03
**Discussion Summary**

As the discussion period ends, we would like to sincerely thank all reviewers for their time and helpful feedback. We are encouraged that the reviewers recognized that our work tackles an important and emerging problem (HSz7, 6gHd), is based on a well-justified mathematical foundation (SDgA), and is validated by broad empirical evaluation (HSz7, SDgA).

Below we summarize the engagement with the reviewers.

- [**Reviewer HSz7**] We clarified the motivation for our formulation, transferability to unseen tasks, and the construction of agent embeddings. We added both conceptual and experimental comparisons to three related works, AttrPrompt (Yu et al. '23), PICLe (Choi et al. '24), and Mixture-of-Personas (Bui et al. '25). The results show that our methods outperform these strong baselines in most cases. We mathematically showed that our methods scale linearly with both the number of agents $M$  and the context size $K$.  The reviewer initially gave an **overall rating of 4**, and could not engage further as the discussion was terminated early on the reviewer's side this year.

- [**Reviewer 6gHd**] We highlighted fundamental differences between our work and that of Baijun et al. '24, showing that our work is novel. We clarified the definition of representation error metric. We compared our work both conceptually and experimentally to AttrPrompt (Yu et al. '23), PICLe (Choi et al. '24), and Mixture-of-Personas (Bui et al. '25). The results show that our methods outperform these strong baselines in most cases. Regarding the survey-based "alignment" concern, experimental analyses indicate that our methods capture meaningful behavioral diversity rather than noises on the OpinionQA dataset. The reviewer confirmed that all of their concerns were addressed and stated that they would give a score increase ("All my concerns have been addressed. I will raise my **overall rating to 4**.")


- [**Reviewer SDgA**] We addressed concerns about using annotations from LLMs in the WikiArt experiment and about the path dependency of our methods. We expanded the discussion of potential future directions, including statistical inference, simulating group dynamics, using advanced agent frameworks with memory and planning, and exploring a single-generalist approach with a SOTA model. The reviewer decided to maintain an **overall rating of 6** ( "I have read the response and decided to maintain my original score.").

We are glad that our responses have addressed the reviewers' comments and concerns. We are grateful for their insightful feedback, which will help us strengthen this work further. We will incorporate these discussions into the final version of our paper.

---

### Meta-Review · Area_Chair_aWQR · 2026-01-03

**Summary:**

The paper proposes selecting a set of in-context–conditioned LLM agents to represent human behavioral diversity, framed as a submodular optimization problem.

Reviewers agree the problem is important, and the empirical evaluation is broad, but they raise substantial concerns regarding conceptual novelty, the motivation for the formulation, and the soundness of key evaluation choices. Much of the approach closely resembles prior submodular example-selection or persona-mixture methods, and several reviewers felt the theoretical contributions are incremental. Additionally, dependence on datasets like OpinionQA, which is known to yield unstable LLM behaviors, and the use of synthetic annotators for WikiArt undermine the strength of the empirical claims. Overall, the submission presents useful engineering but falls short of the conceptual and methodological rigor expected for acceptance.

**Reviewer Concerns:**

The rebuttal effectively clarified embedding construction, added missing baselines (PICLe, AttrPrompt, Mixture-of-Personas), and extended the analysis of scalability and theoretical framing. However, the main unresolved issues remain central: (1) the optimization objective ignores population distribution, limiting relevance for tasks requiring statistical fidelity; (2) the sequential greedy construction introduces path-dependence that weakens the purported guarantees; (3) the OpinionQA evaluation relies on a dataset with known methodological flaws; and (4) the WikiArt study uses LLM-generated “synthetic humans,” raising doubts about whether the method actually captures real human behavior. These concerns leave the core claims insufficiently supported despite the thoughtful rebuttal additions.

**Reviewer Scores:**

Reviewer HSz7: Likely unchanged (4), as conceptual-motivation issues remain despite clarifications.

Reviewer 6gHd: Final rating moved to (2 -> 4), but still views the contribution as incremental.

Reviewer SDgA: Unchanged (6), supportive but acknowledges major limitations.

---

### Decision · Program_Chairs · 2026-01-26

Reject